# Differentiating between Natural and Modified Cellulosic Fibres Using ATR-FTIR Spectroscopy

**Ludovico Geminiani** [1,2,*], **Francesco Paolo Campione** [2,3,4], **Cristina Corti** [2,3], **Moira Luraschi** [2,4], **Sila Motella** [2,5,6], **Sandro Recchia** [1] **and Laura Rampazzi** [2,3,7]

1   Dipartimento di Scienza e Alta Tecnologia, Università degli Studi dell'Insubria, Via Valleggio 11, 22100 Como, Italy
2   Centro Speciale di Scienze e Simbolica dei Beni Culturali, Università degli Studi dell'Insubria, Via Sant'Abbondio 12, 22100 Como, Italy
3   Dipartimento di Scienze Umane e dell'Innovazione per il Territorio, Università degli Studi dell'Insubria, Via Sant'Abbondio 12, 22100 Como, Italy
4   Museo delle Culture, Villa Malpensata, Riva Antonio Caccia 5, 6900 Lugano, Switzerland
5   Dipartimento di Storia, Archeologia e Storia dell'Arte, Università Cattolica del Sacro Cuore, Largo Gemelli 1, 20123 Milano, Italy
6   Laboratorio di Archeobiologia, Musei Civici di Como, Piazza Medaglie d'Oro 1, 22100 Como, Italy
7   Istituto per le Scienze del Patrimonio Culturale, Consiglio Nazionale delle Ricerche (ISPC-CNR), Via Cozzi 53, 20125 Milano, Italy
*   Correspondence: lgeminiani@uninsubria.it; Tel.: +39-031-238-6475

**Abstract:** This paper presents the limitations and potential of ATR-FTIR spectroscopy applied to the study of cellulosic textile collections The technique helps to differentiate natural fibres according to the content of lignin, pectin, hemicellulose, and wax, although some problematic issues should be considered. The spectral differences derived from the environmental humidity uptake and the plant composition are reviewed and discussed in the light of new experimental data. Diagnostic bands are proposed that can discriminate between different fibres from different plants. The contribution of ageing is also considered, demonstrating that sometimes aged fibres cannot be reliably recognised. In contrast, the potential of ATR-FTIR spectroscopy to discriminate between natural and modified fibres is discussed and proven. The best results were obtained when microinvasive ATR-FTIR spectroscopy was coupled with SEM observations. The proposed protocol was tested on microsamples of various cellulosic materials from traditional Japanese samurai armours dating from the 16th to the 20th centuries (Morigi Collection, Museo delle Culture, Lugano, Switzerland). The results facilitated a complete characterisation of the materials and demonstrated that the protocol can be used to study a wide variety of cellulosic materials, including both natural and man-modified fibres, and paper.

**Keywords:** cellulose; ATR-FTIR; SEM; viscose; natural fibres; paper; ageing

## 1. Introduction

Vegetable fibres for textile materials were first used in prehistoric times [1,2]. Hemp is probably the oldest cultivated plant, which was widespread from Southeast Asia to China, where it seems have been used since around 4500 BCE. Several places of origin of the plant have been proposed, but probably archaeological findings in different centres are indicators of the diffusion of the plant in the early human agriculture [3]. Similarly, flax was certainly cultivated by the 4000 BCE in the Egyptian area, although it seems to have originated in the Near East [4,5]. Lastly, the art of spinning cotton appeared in India from around 3000 BCE, but it also developed independently in Peru [4].

Vegetable fibres are divided into groups based on their origin within the plant. Cotton originates as hair on the seeds, where each fibre consists of a single, long, narrow cell. Flax, hemp, jute, and ramie are bast fibres in the inner bast tissue of stems and are made up of overlapping cells. Chemically, all these fibres mainly consist of cellulose, although

their cell walls also contain varying amounts of different substances such as hemicellulose, lignin, pectins, and waxes, which can be removed or reduced by processing [2,5,6]. While hemicellulose [7] is chemically similar to cellulose, pectins [7] are acidic polysaccharides mainly composed of homogalacturonan, hamnogalacturonan I and II, and different neutral sugar side chains. Waxes [8] are located in epidermal cells in the plant cuticle and are made up of long-chain aliphatics, such as alkanes, alcohols, aldehydes, fatty acids, and esters, mixed together with varying amounts of cyclic compounds such as triterpenoids.

Paper is not strictly a natural fibre; however, its constituents are natural fibres, especially if it is ancient paper [9]. In the West, paper was made from flax and cotton rags or a mixture of these second-hand fibres, whereas in the East plant sources were more common.

The raw materials used to produce such fibres include bast plants, tree bark, grass stalks, and other vegetation. In China, hemp and mulberry were the most ancient sources, followed by rattan, whose reserves were exhausted in the 12th century. Bamboo (*Bambusoideae* subfamily) and mulberry (*Morus alba*) or paper mulberry (*Broussonetia papyrifera*) [10], thus replaced other plants as the main sources of fibre for papermaking. For special kinds of paper, rice, and wheat straw [11], the bark of sandalwood (*Dalbergia hupean*) and other trees, hibiscus stalks (*Hibiscus mutabili*), seaweed, and certain other plants were also used.

In Japan, besides bamboo, other raw materials were used [12], such as *kozo* (Japanese name for mulberry), *ganpi* (*Wikstroemia* spp.), and *mitsumata* (*Edgeworthia papyrifera*) plants. With the start of industrial papermaking, since the beginning of the 19th century, wood pulp has become the chief raw material, which gradually replaced all other sources. This kind of paper presents particular conservation problems, mainly due to the acidic pH derived from the processes and substances used in the industrial method [13], while ancient papers generally decay as do the constituting fibres [10].

The end of the 19th century saw the emergence of chemically modified and then regenerated cellulosic fibres, which were developed both to reuse waste from cotton production and to design fibres with specific desirable properties [1]. The first commercially produced man-made fibre [14] was Chardonnet silk, which from 1891 was obtained from nitrocellulose. One year before, in 1890, the French chemist Louis-Henri Despeissis patented a process for making fibres from cuprammonium rayon to obtain the so-called Bemberg silk (from the name of the German textile firm who first started the commercial production).

Today, the most commonly used type is viscose rayon (or simply viscose), produced since 1905 by the British silk firm Samuel Courtauld & Company as a substitute for silk. A more environmentally friendly type of rayon was developed in 1988 and sold as Lyocell. These are all defined as regenerated fibres [14] because the cellulose, obtained from soft wood or from the short fibres (linters) that adhere to cottonseeds, is converted into a liquid compound, squeezed through tiny holes in a device called a 'spinnerette', and then converted back to cellulose in the form of fibre. The main advantage is that the lustre, strength, elongation, filament size, and cross section can be controlled. Rayon's properties are similar to those of cotton, but its aesthetic characteristics can be used to make it resemble silk. The purified cellulose is first treated with a sodium hydroxide solution. After the alkali cellulose has aged, carbon disulphide is added to form cellulose xanthate, which is dissolved in sodium hydroxide. Alternatively, cellulose can be dissolved in a solution of copper salts and ammonia (cuprammonium rayon) or in a nontoxic amine oxide solvent (Lyocell).

The most serious conservation problem for cellulosic materials is the photochemical damage [10,15]; however, oxidation can occur in a dark and humid environment, especially when there are temperature fluctuations [10,16,17]. After absorbing electromagnetic radiation, free radical photochemical reactions start in the cellulose. Reactive oxygen species (ROS) are produced by photochemical reactions, which are catalysed by the presence of transition metal ions—generally coming from mordants and dyes—and are accelerated by other factors such as moisture. The product of the oxidation is called 'oxycellulose'. However, hemicellulose appears to act as a sacrificial substrate, which oxidates in lieu of

cellulose [18], thus fibres containing hemicellulose can benefit from extra-protection against UV light damage.

Lignin probably also acts as a radical scavenger [17–20], despite it being considered to cause damage. The literature [15,17] also stresses that controlling humidity levels is essential in the conservation of paper and of both modified and natural cellulosic textiles. Too high levels can lead to a microbiological attack as well as migration of dyes and pigments, while shrinkage and brittleness are caused by too low levels. Even the speed of drying is an important factor, as it influences the amount of bound water remaining in the polymers of the fibres after drying. If the speed of evaporation is too high, effects similar to desiccation can occur. On the other hand, rayon viscose is weaker when it is wet, and so it is fundamental that mechanical support is provided during the wet cleaning processes.

Many analytical studies on cellulosic fibres have been carried out. Their industrial use has promoted studies on crystallinity and water sorption of both natural and regenerated cellulose since the late 1940s [21]. More recently, the issue of water interaction with cellulose has been investigated, especially with thermal analyses [22,23]. ATR-FTIR spectroscopy has been used to quantify the water uptake, coupled with chemometrics [24]. It has also been used to investigate different materials such as paper [25] and epoxy resin [26].

The hygroscopic behaviour of cellulosic plants was reviewed by Celino [27], who studied flax, hemp, jute, and sisal [28]. The crystallinity index of the fibres is usually investigated by X-ray diffraction (XRD) [11,23,28–30]. The degree of polymerisation can be tested with size exclusion chromatography (SEC) [19], while the oxidation progress can be examined by FTIR spectroscopy [16,19], UV–Vis spectroscopy [10,16,19], fluorescence spectroscopy [10], surface pH measurements [10], and titrimetric methods [10].

When the fibres need to be clearly identified, several methods can be used [31,32]. Microscopy has traditionally been used to differentiate fibres, using light microscopy (LM) [33,34], polarised light microscopy [12,33,35], and scanning electron microscopy (SEM) [33,34,36,37]. While for cotton, for example, the identification is quite simple, some fibres can be easily confused due to the high natural variability of descriptive parameters. This is true especially for bast fibres [36]. Therefore, considerable experience is required [35]. Stain tests are often used as a complementary assay to confirm results obtained from other examinations or for special analyses [15]. For example, they are used to evaluate differences in the lignification—that is, to detect lignin content—using the Herzberg stain [38]. Anyway, stain tests may fail with dark-coloured archaeological or dyed fibres [15] and they are considered destructive [31]. Today, Raman [34,39] and FTIR spectroscopy [34,40–43] are established techniques to identify the nature of the fibres, both cellulosic and proteinaceous, generally coupled with SEM observation. In particular, ATR-FTIR spectroscopy is used to give chemical information about archaeological textiles, whose morphology is often very decayed due to the biological attack [33,40,41,44–46]. ATR-FTIR analysis permits also to detect a small amount [15] of dyes, mordants, contaminants, and dirt [47,48] such as gypsum, kaolin, and various organic materials which can adhere to historical textiles [45] and papers [49].

Generally speaking, FTIR can distinguish fibres from different herbaceous plants too, but problems arise when bast fibres are aged [33]. Garside's study [50] on flax, hemp, jute, ramie, cotton, and sisal proposed the use of two ratios based on the intensities of infrared bands representing the overall organic content (2900 cm$^{-1}$), lignin content (1595 cm$^{-1}$), and cellulose content (1105 cm$^{-1}$). These ratios were used to differentiate fibre types, however some problems remained, especially due to the influence of relative humidity. Garside [51] then proposed polarised ATR-FTIR spectroscopy and microspectroscopy to assess the differentiation between flax and hemp. Although the technique also obtains indications of the total cellulose crystallinity of the material [52], data collection and interpretation are more complicated than the non-polarised ATR-FTIR mode. Similarly, good results are obtained when ATR-FTIR data are processed with the chemometrics method in order to cluster fresh fibres from different sources [33,53]. However, the best performances are obtained when advanced chemometric methods are applied [54]. FTIR investigations

into flax [55] have provided very detailed results which can estimate, for example, how fine the flax fibre is. Similarly, ATR-FTIR spectroscopy is particularly efficient [56,57] in discriminating between cellulose I, II, and III obtained by processing cotton with NaOH solution and liquified anhydrous $NH_3$, respectively.

Finally, it is worth recalling advantages of analysis with ATR-FTIR spectroscopy [58]:

— no sample preparation;
— no band saturation phenomena;
— time and cost saving;
— few micrograms or less are generally necessary;
— non-destructive, as the sample can be re-used for further investigations (although the pressure applied during the analysis can induce morphological modifications);
— extensive database available, as literature dealing with transmission FTIR can be generally extended to ATR spectra.

## 1.1. Aim of the Study

Our work derived from the need to analyse microsamples from a collection of traditional Japanese samurai armours from the 16th to 20th centuries (Morigi Collection, Museo delle Culture, Lugano, Switzerland). The aim was to obtain the maximum amount of information from the materials under analysis. The results are a part of a challenging characterisation of the majority of materials in the armours, which has never been done before. A mixed protocol based on ATR-FTIR spectroscopy and SEM observation was adopted. Non-destructive analyses were performed so that future analyses could also be carried out, for example on dyes. Although FTIR spectroscopy has been widely tested for the investigation on fibres, we unexpectedly found that the literature lacks an overview for those approaching the topic for the first time. In addition, the wide bibliography on cellulose makes it quite difficult to verify the causes of the variability of FTIR spectra.

The aim of this paper Is thus to discuss the potential and limitations of ATR-FTIR spectroscopy for studying cellulosic fibres. In addition, we revised ATR-FTIR assignments for the different components of cellulosic fibres (cellulose, lignin, hemicellulose, pectins, waxes) in order to highlight spectral differences. Using UV-accelerated ageing tests, the different influence of the decay on the components was also considered. Finally, we investigated the water adsorption by completely reviewing the band assignation for cellulose I and II, and by studying cotton and viscose spectra under low and high humidity conditions. Spectral deconvolution was employed to also visualise the contribution of the water OH stretching band to the total band. A protocol is also proposed for the analytical investigation of cellulosic fibres in paper and books, as well as in textile collections.

## 1.2. Theoretical Background

From a chemical point of view [15], the main difference between cotton and viscose is the partial degradation of cellulose during viscose production. Cellulosic fibres contain both crystalline and amorphous domains. In comparison with cotton, where the crystalline parts account for 70–75% of the fibre, in viscose, the proportion of crystalline regions is only 25–30%. In addition, due to alkaline treatment, the crystal structure of native cellulose (cellulose I) turns into that of cellulose II. The hydrogen-bonding pattern changes accordingly [57,59] (Figure 1). The increased proportion of the amorphous phase also gives viscose a higher moisture regain [15]. Polar solvents such as water can easily form extra hydrogen bonds with cellulose, as some oxygen atoms in the chain are non-hydrogen bonded proton acceptor [60] (Figure 1).

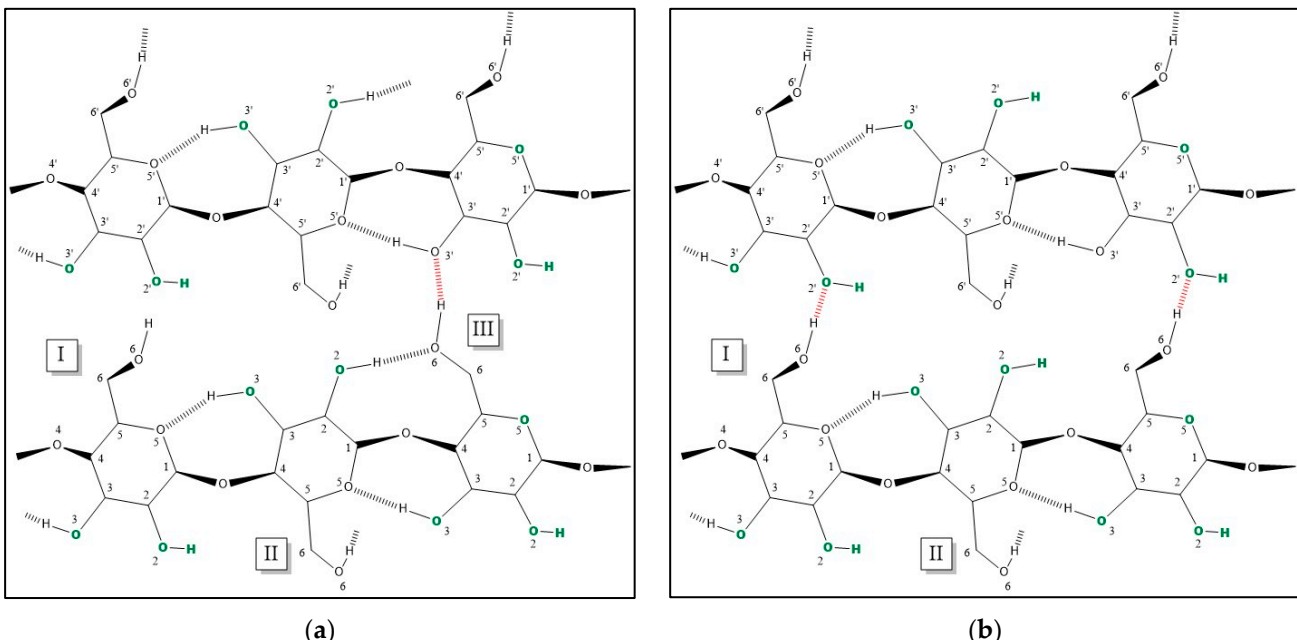

(**a**)                                                      (**b**)

**Figure 1.** Hydrogen bonding pattern in cellulose I (**a**) and II (**b**). For both, 020 plane is represented. Green-coloured atoms are thought to bind water molecules. Intermolecular bonds are highlighted in red. In cellulose II, $O(2)H\cdots\cdots O(2')$ hydrogen bond is not shown as it is visible on another plane (110) [57]. Roman numbers in labels stand for different conformations which C(6)O(6) group can assume in the same chain [59].

However, water and chemical agents can only access the amorphous regions and the surface of the crystalline regions [15]. In fact, in the amorphous regions intermolecular cross-bonds continually break and re-form owing to the thermal vibrations, preferably re-forming bonds with water molecules instead of cellulose units [61,62]. This enables water to penetrate by capillarity, disrupting the interchain bond ($6OH\cdots\cdots O3'$), which is typical of crystalline regions. According to the literature [61–63], water molecules can only bind oxygen atoms which are in positions 2, 3, 5 (green-coloured in Figure 1). However, it should be considered that the absorption of water is controlled by the accessibility of the hydrogen bonds to moisture, which strongly depends on the crystallinity degree [64].

According to the environmental humidity, three types of water can be found [15], showing different bond strengths: (i) structural water; (ii) bound water; (iii) excess water. The latter can be easily eliminated by centrifuge or pressure. Structural and bound water penetrate cotton deeply, whereas they act as plasticisers for the fibre. As a consequence, regenerated fibres are more plastic than natural ones, due to the increased water uptake. This fact makes handling wet viscose textiles quite dangerous because they can readily stretch and lose their shape [15]. Structural water forms a monolayer and is directly bound to hydroxyl groups of cellulose. It is thus difficult to eliminate, and it is known not to freeze, even below 0 °C [22,65]. For this reason, it is also called 'non-freezing water'. Bound water derives from the moisture that is adsorbed by the fibre and is loosely bound to cellulose, and thus can evaporate by heating or by keeping the material in a dry environment. Under 0 °C, this kind of water is ice, and thus it is also called 'freezing bound' [22,65].

## 2. Materials and Methods

### 2.1. Reference Materials

Modern samples of drawing paper (Fabriano, weight 180 g/m²), pinewood, cotton, jute, flax, and viscose were purchased from specialised shops. Historical sources report that any chemical was used in the processing of hemp [66]. For this reason, we chose to test non-woven fibre (hemp thread, made in Egypt, distributed by GoPlast s.r.l, Varese, Italy), which is probably philologically nearer to the material used in the past. Japanese

paper (weight 15 g/m$^2$) was obtained from a mixture of *kozo* and Manila hemp fibres, as specified by the seller (CTS Conservation-Milan). Naturally aged samples of flax came from a private collection and date back to the beginning of the 20th century. Accelerated ageing was conducted with a 100 W UV led lamp (producer: Everbeam). The emission spectrum was measured, showing a band width of nearly 30 nm centred at 365 nm. As cellulose absorbs mainly in the UV region [15], in order to evaluate the effects of indoor ageing, it was only used to test UV-A radiation [67]. To control the water uptake by the fibres, a laboratory ventilated oven was used to reproduce dry conditions (65 h, 40 °C). High humidity conditions were obtained by storing samples for 65 h in a desiccator over K$_2$SO$_4$ salt, where RH level was monitored in situ by placing moisture data logger and set to around 97%. When not otherwise specified, samples were analysed under laboratory environmental conditions.

## 2.2. The Morigi Collection of Traditional Japanese Armours

Cellulosic samples studied in this research came from seven Japanese full armours, which are part of the Morigi Collection. This collection was donated to the Museo delle Culture Lugano (MUSEC) in 2017 by the collector Paolo Morigi, who already owned one Japanese armour (2017.Mor.4) and acquired all the others in two auctions held in Nice and Paris in June 2016. The collection was presented in a temporary exhibition at the MUSEC in 2018 and is now on permanent display.

The collection includes different styles and dates of the Japanese armours.

All the armours are *kinsei gusoku* ('modern time armour'), and date back to different historical ages: the Azuchi-Momoyama period, the harshest period of feudal wars in Japan spanning the second half of the 16th century; the peaceful Edo period (1603–1868); the Meiji period (1868–1912); the Taishō period (1912–1926); and the Shōwa period (1926–1989). Some (2017.Mor.1 and 2017.Mor.7) are battle armours of the Azuchi-Momoyama period which were made to be used on the battlefield, so they are anatomically shaped, and comfortable to wear. The others were made when armours were only used for celebrations and parades. Armour 2017.Mor.9 is a *kinsei gusoku* made according to the old-fashioned style (*mukashi gusoku*, 'once upon a time armour') used in the Middle Ages.

The main features of the armours from which each sample comes are summarised in Table 1. The microsamples are shown in Figure 2 and listed in Table 1. The pictures of the whole historical objects from which samples were taken are shown in Figure S1.

**Table 1.** Microsamples from the Morigi collection.

| Armour | Microsamples | Armour Type | Presumed Dating |
|---|---|---|---|
| 2017.Mor.1 | 1_11 | *kinsei gusoku* | late 16th |
| 2017.Mor.5 | 5_3 | *kinsei gusoku* | 20th |
| 2017.Mor.6 | 6_1 | *kinsei gusoku* | early 20th |
| 2017.Mor.7 | 7_6a, 7_6b | *kinsei gusoku* | 17th |
| 2017.Mor.8 | 8_1 | *kinsei gusoku* | 17th |
| 2017.Mor.8 | 8_9 | *kinsei gusoku* | early 19th |
| 2017.Mor.9 | 9_6 | *mukashi gusoku* | late 19th |
| 2017.Mor.10 | 10_1 | *kinsei gusoku* | early 20th |

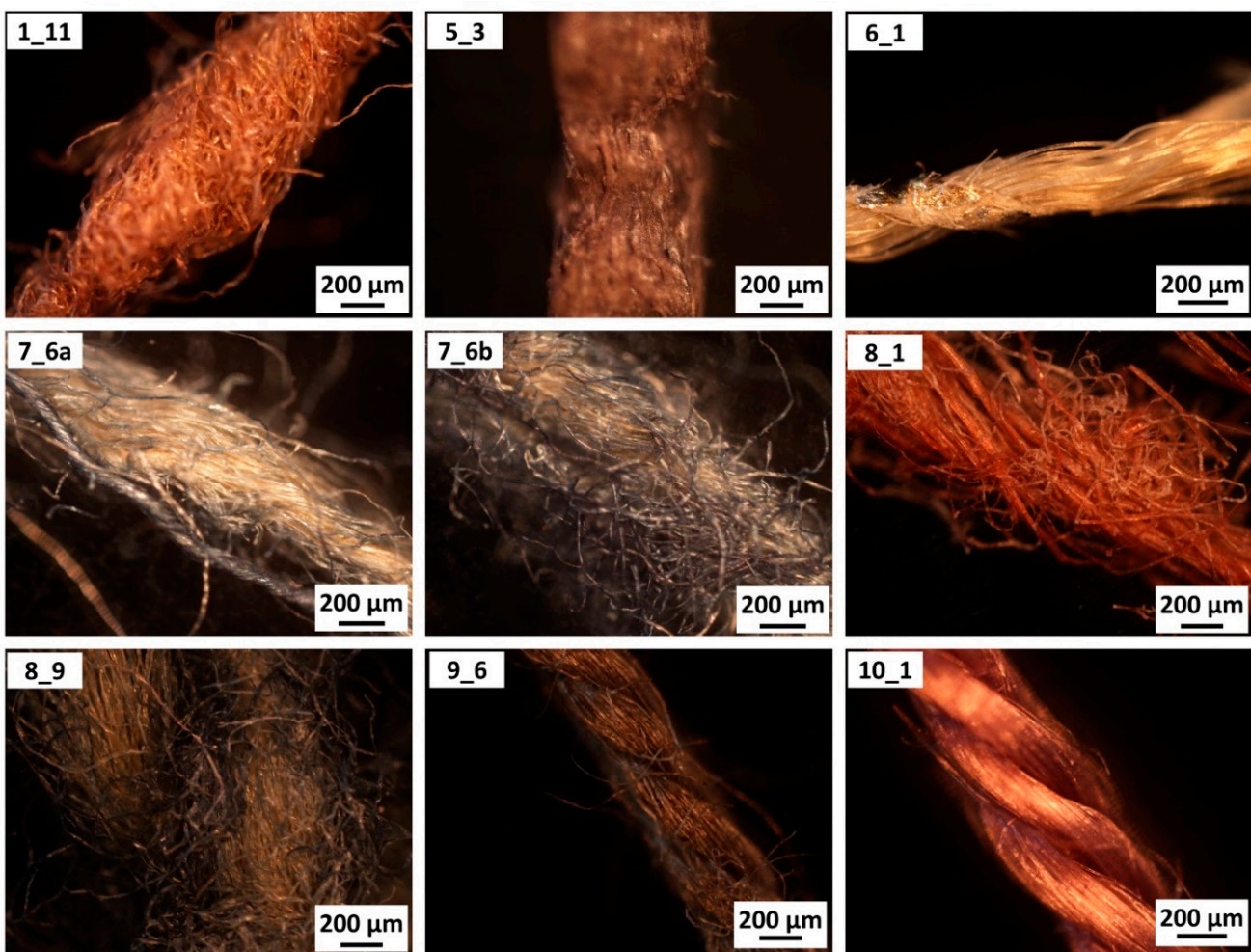

**Figure 2.** Optical microscopy images of analysed samples from the armours.

### 2.3. Sampling

The samples, generally only a few millimetres long, were taken from previously damaged areas or in hidden places. When it was necessary, we made use of digital microscopy during the sampling to ensure the homogeneity of the sample. The samples were collected using scissors and tweezers and stored in LDPE containers until analysis in the laboratory.

### 2.4. Optical Microscopy

The textiles from which the samples were taken were observed in situ with a portable digital microscope MAOZUA USB001 (Mustech Electronics Co., Ltd., Shenzhen, China) and images were acquired using the software MicroCapture Plus, version 3.1.

The thread samples were then observed in the laboratory with an optical microscope Nikon Eclipse LV150 (Nikon Corporation, Tokyo, Japan), equipped with a Nikon DS-FI1 digital image acquisition system. Images were acquired and elaborated using the NIS-elements F software, version 3.22.

### 2.5. Scanning Electron Microscopy (SEM)

The samples were observed without any pre-treatment with a FEI/Philips XL30 ESEM (low vacuum mode—1 torr, 20 kV, BSE detector) (FEI Company, Hillsboro, OR, USA).

### 2.6. Attenuated Total Reflectance Fourier Transform Infrared Spectroscopy

ATR-FTIR spectra were acquired by means of a Thermo Scientific Nicolet iS10 instrument, in the range between 4000 and 600 $cm^{-1}$, 4 $cm^{-1}$ resolution, 32 scans. The detector is a fast recovery deuterated triglycine sulphate (DTGS). The analysing crystal is diamond, which shows typical absorption at around 2100 $cm^{-1}$. For this reason, in figures, the region 2400–1800 $cm^{-1}$ is generally not shown. The background was periodically acquired to permit to the software the automatic subtraction of atmospheric air spectrum from the sample spectrum.

### 2.7. Data Treatment and Elaboration

Spectra were interpreted by comparison with a homemade reference database and with the literature. Spectragryph optical spectroscopy software, Version 1.2.16.1, was used to visualise and manipulate ATR-FTIR spectra [68].

The baseline correction was applied to all the spectra. As FTIR spectra are influenced by scattering effects, it is a common practice to remove their influence from spectra through pre-processing methods. The standard normal variate (SNV) approach effectively removes the multiplicative interferences of scatter and particle size [69]. For this reason, SNV pre-processing was applied either to the whole spectrum or to a part of it, when needed. It should be considered that this kind of data elaboration excludes information about absolute intensity. On the other hand, the fine differences in the band shape of different superimposed spectra are enhanced.

The application of SNV is based on the following mathematical operation:

$$y_{ij \, (SNV)} = \frac{y_{ij} - \overline{y}}{\sqrt{\frac{\sum (y_i - \overline{y})^2}{n-1}}} \tag{1}$$

that is subtracting the mean spectra $\overline{y}$ to each intensity value $y_i$ of the original spectrum and then dividing for standard deviation value.

### 2.8. Spectral Analysis and Curve-Fitting

The O-H stretching band was analysed by a band fitting method. First of all, selected spectra were truncated down to the 4000–2980 $cm^{-1}$ range (O-H stretching region), and baseline correction was applied using a linear function passing through the ordinates at the endpoints of the interval considered. Band fitting was performed using the Multiple Peak Fit function of the peak analyser package of Origin Pro 2018 software (version SR1 b9.5.1.195, OriginLab corporation, Northampton, MA, USA). First, the second derivative of the convoluted spectra was smoothed by the adjacent-averaging method (smoothing window size of 20) and used to locate the band positions. Then, the spectra were deconvoluted using Voigt curves and a constant baseline (constrained at zero absorbance). Bands were allowed to move from their initial position in a range reported in the literature (refer to Results and Discussion section), while the full width at half height (FWHH) of the bands was fit by the program in a range from 0 to 100 $cm^{-1}$ for cellulose OH stretching, and from 0 to 200 $cm^{-1}$ for water OH stretching. The fitting was iterated until convergence, and a Chi-Sqr tolerance value of $10^{-6}$ was reached.

The Voigt curve (which is a convolution of a Gaussian function and a Lorentzian function) was chosen instead of a pure Gaussian or Lorentzian, as it helps to take into account broadening effects which are typical of hydrogen bonded mixtures [70]. The tight network of hydrogen bonding also acts like a discharge path for the vibrational energy, and the more rapid the loss of excitation, the broader the resulting peak. On the other hand, a long lifetime leads to narrow peaks (as happens in the gas phase) [71]. Unlike previous studies [57,61], we included water bands (refer to Result and Discussion section), as in our opinion it is important to also consider contributions from OH stretching of bound water [25] to the overall band. In fact, a large band width is expected for OH stretching in water [71], but this is not the case for H-bonded OH stretching in cellulose, which shows specific absorptions [63,72] in a limited range.

## 3. Results and Discussion

### 3.1. Variability of Natural Cellulose Fibres

In order to test their origin, the various natural fibres based on cellulose were observed with SEM (Figure 3a–d). Red arrows highlight morphological features which help the identification task. Literature is rich of morphological studies on natural fibre [5,36,38,73].

Depending on its origin, **cotton** shows marked differences in external characters, both in colour (ranging from white to yellowish to reddish-brown) and fibre length (varying from 19 to 60 mm). Under the microscope, the fibre appears as single-celled, ribbon-like with spiral circumvolutions (Figure 3a) and with two terminations (not visible in the shown image), one spatula-like, the other torn and defibred. Where the cross-section is visible, fibres with an oval or elliptical outline can be seen, more or less flattened, never circular; their diameter varies between 15 and 35 μm.

The individual fibres of **jute** are 2 to 5 mm long, with a diameter of 20 to 25 μm. On microscopic examination, the fibres appear grouped in bundles (Figure 3b), held together by pectic and encrusting substances. When visible, single sections appear polygonal, with an oval or spherical lumen. The fibres have few longitudinal striations and irregular walls of varying thickness.

The individual fibres of **flax** vary in size between 6 and 50 mm; the diameter is between 10 and 40 μm. The individual fibres appear cylindrical in shape, with a thin central channel, and terminate with a sharp tip as already mentioned (not visible in the image shown). Looking at the fibres longitudinally, marked transverse striations are observed, giving the filament the appearance of a bamboo cane (Figure 3d). When observed in cross-section, the flax exhibits irregular, polygonal contours with a central spot corresponding to the lumen.

**Hemp** fibre consists of bundles of free-standing filaments, 30 to 70 cm long; the individual fibres are 15 to 50 mm long, with a diameter varying between 15 and 35 μm. Under the microscope, the fibre is made up of slightly flattened cylindrical fibres. As in flax, transverse, often crossed striations are also present, resembling the knots of a bamboo cane (Figure 3c). The fibre walls are irregularly thick, with a central channel thinning towards the end. The termination (not visible in Figure 3c) is spatula-like, rounded, unlike flax where it is often sharp. When visible, the cross-sections show an irregular outline, with a broad, star-shaped lumen. A differentiation of hemp from flax under the microscope is not easy, as their morphology is similar. Hemp, however, has a more irregular fibre shape, more flattened and with more knots, as well as a rounded termination.

As far as **paper** is concerned, its appearance under SEM shows great variability depending on the fibres from which it is made. In general, it appears as a mixture of disorderedly arranged fibres, with a high number of inorganic particles both in the space between the fibres and on the fibres themselves [74].

A sequential procedure which helps to use properly SEM observation is resumed in Figure 4a.

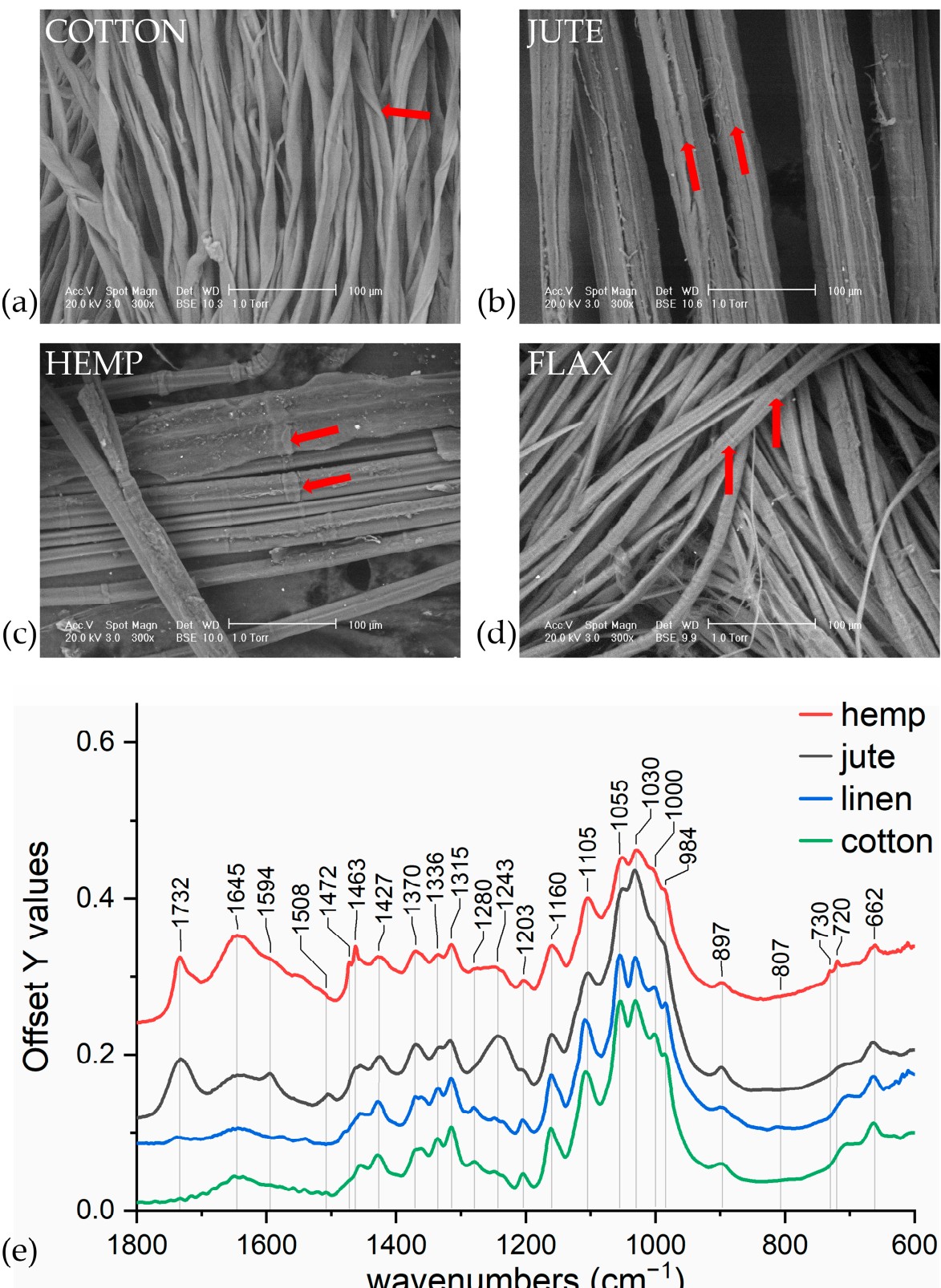

**Figure 3.** (**a**) SEM image of reference cotton; (**b**) jute; (**c**) hemp; (**d**) flax; (**e**) detail of ATR-FTIR spectra of reference materials of jute, hemp, flax (linen), and cotton (region 1800–600 cm$^{-1}$). Red arrows highlight morphological features which help the identification task.

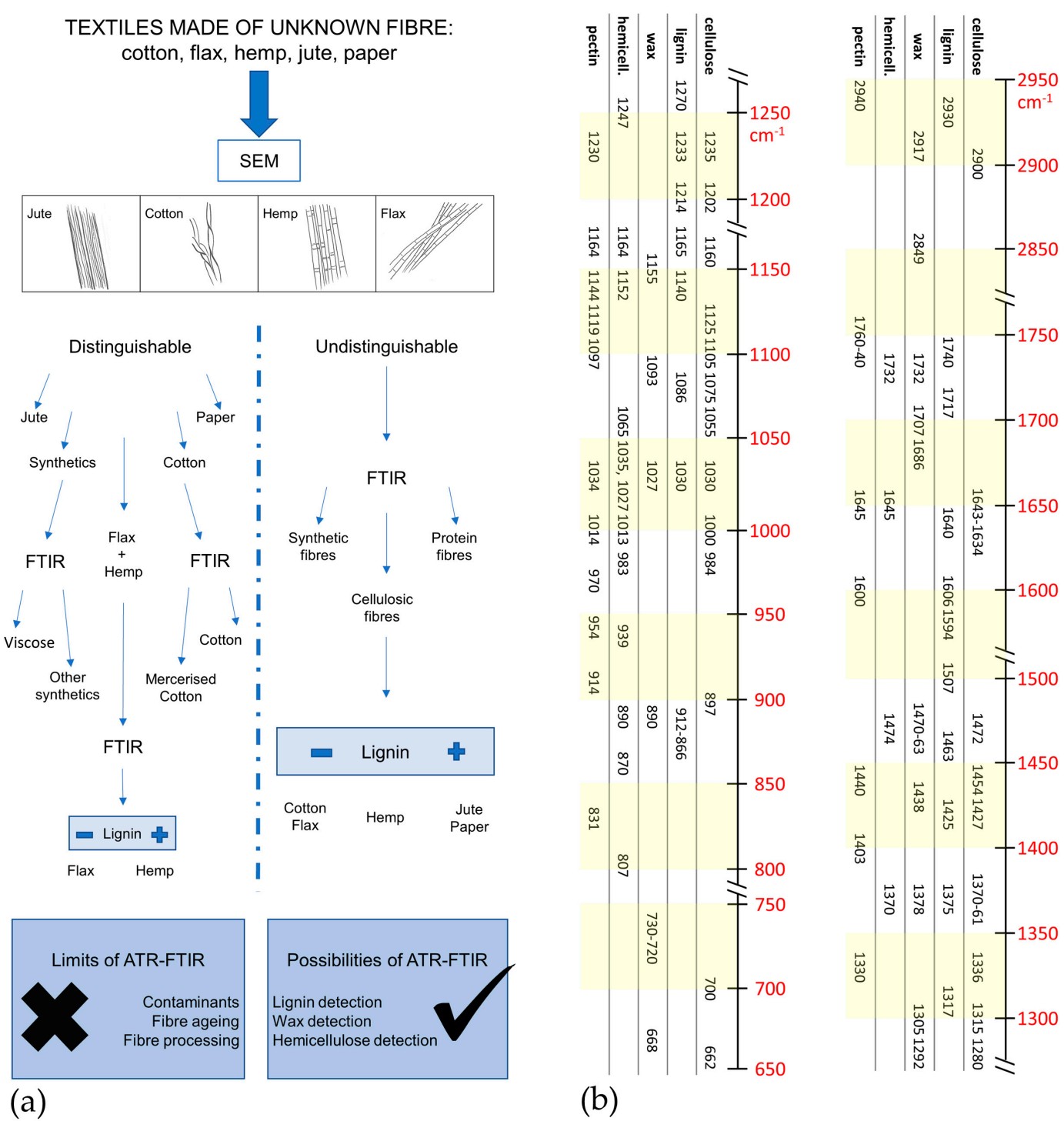

**Figure 4.** (**a**) Sequential procedure to differentiate cellulose fibres; (**b**) Spectral features associated to cellulose and to correlated compounds (water, lignin, pectin, hemicellulose, wax). The complete assignations are exposed in Table S1.

The same fibres were tested with ATR-FTIR spectroscopy (Figure 3e). The whole spectra are shown in Figure S2. In line with the literature, they all showed the main spectral features of cellulose, which are presented in Figure 4b. Some of them also present signals from other natural compound such as lignin, hemicellulose, wax, and pectin [8,50,55], whose spectral features are reported in Figure 4b as well. The complete assignations, which are exposed in Table S1, were obtained from literature search [7,8,50,55,57,59,75–85]. A sequential procedure which helps to use properly SEM observation coupled with ATR-FTIR analysis is resumed in Figure 4a. Figure 4b should be used as a easy-to-use table for differentiating signals from cellulose, lignin, pectin, hemicellulose and wax.

The principal differences are due to the presence of other natural compounds. It is evident that they share many bands, but specific features were highlighted and proposed for each group. In particular:

- for lignin: 2920, 1740, 1717, 1606, 1594, 1507 $cm^{-1}$;
- for wax: 2926, 2854, 1732, 1707, 1686, 730–20 $cm^{-1}$;
- for hemicellulose: 1247, 1013, 939, and 807 $cm^{-1}$;
- for pectin: 1740, 1600, 1141, 1097, 1014, 954, 914, and 831 $cm^{-1}$.

The co-presence of these different compounds in each plant explains the complexity of natural plant spectra. However, the fresh fibre from different plants can be differentiated quite easily [50]. Bast fibres, that is all except cotton, should show the particular lignin peaks at 1594 and 1507 $cm^{-1}$. As FTIR is highly sensitive to lignin, it could be considered a non-destructive alternative for Herzberg stain test [38], which is time and reagent consuming.

Jute contains much more lignin than the others [50], and thus shows clearer lignin peaks. Similarly, the peak at 1735 $cm^{-1}$ should be assigned to C=O of ester in pectin, wax, and hemicellulose.

Hemp is the only one characterised by wax signals at around 1735, 1470, 730, and 720 $cm^{-1}$. Long-chain fatty acids are the most abundant lipidic fraction [86]. However, wax signals can disappear due to the fibre processing [87–90]. It also shows signals that can be attributed specifically to pectin (around 1600, 955 $cm^{-1}$) and to proteins [7,91], that is the enhancement of the band at around 1650 $cm^{-1}$ and the peak at 1550 $cm^{-1}$.

Flax spectrum resembles cotton more than other bast fibres. Lignin signals do not appear, while the peak at 1730 $cm^{-1}$ could be interpreted as the presence of hemicellulose and pectin, which hardly appear in cotton. Similarly, the peak at 807 $cm^{-1}$ is proof of the presence of hemicellulose, whose flax is rich [5]. In fact, the percentage lignin content is much lower than for hemp and jute, while hemicellulose content is quite higher [2,5,6,50]. However, ATR-FTIR limitations in differentiating bast fibres cannot be ignored [33]. As proof, in Figure 5, we report spectra of different historical samples of flax which were previously identified by SEM observations (not shown).

Samples l1 and l3 appear as pure cellulose fibre, with no signals from lignin or pectin. At first sight, spectra from samples l2 and l4 show peaks at 1740 $cm^{-1}$, which could be attributed to pectin or hemicellulose, and 1576 $cm^{-1}$ which seems to be proof of the presence of lignin. However, the peak at 1735–40 $cm^{-1}$ is a complex band [16,55] assigned to C=O stretching vibration of acetyl or carboxylic groups, which are attributable to pectin, hemicellulose, but also to degradation products of lignin and cellulose (oxycellulose) [16,50]. Similarly, the peak at 1576 $cm^{-1}$ cannot be assigned to lignin (absorption at 1595 $cm^{-1}$) and is probably related to cellulose oxidation [16]. Consequently, the differences in the region 1800–1500 $cm^{-1}$ are not actually differentiating between samples of flax and pure cellulosic fibre, but rather the state of decay of the flax fibre [33].

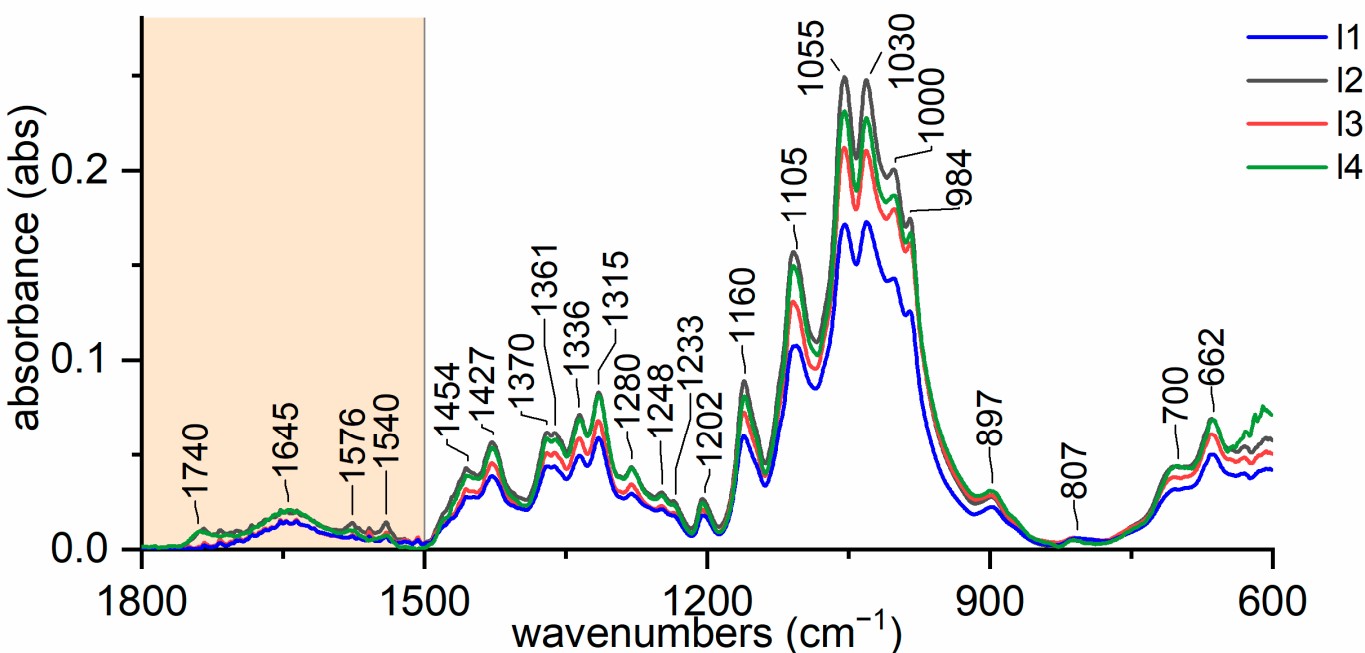

**Figure 5.** Detail (region 1800–600 cm$^{-1}$) of ATR-FTIR spectra of various samples of flax.

In addition, the presence of contaminants [15,47,48] and the processing of the fibre [55] could have some influence on the spectra. It is known that FTIR signal is influenced by the treatment of natural fibres with different reagents [87,89], even if historical methods are not reported to use chemicals [5,66]. For example, alkalisation with sodium hydroxide removes materials, such as hemicellulose, lignin, wax, and pectin, from the surface of fibre bundles [88]. Figure 6 shows an example.

Figure 6a shows the spectra of commercial drawing paper, Japanese paper, and Japanese paper exposed to UV ageing for 60 h. Japanese paper appears different from commercial drawing paper. In fact, Japanese paper shows an increased water uptake (adsorbed water at 1640 cm$^{-1}$), hemicellulose (due to the peak at 1730 cm$^{-1}$), and probably also lignin due to the peak at 1592 cm$^{-1}$. The difference presumably stems from the production process. Commercial drawing paper is produced by the Kraft process, which extracts cellulose from wood pulp. This explains the complete absence of lignin and hemicellulose. During the Kraft process, fillers are added, such as clay or limestone [13], which explains the presence of calcium carbonate (which appears at 875 and 711 cm$^{-1}$) [92]. Japanese paper, in contrast, is obtained from the bark of *kozo* and from the stem of Manila hemp, which are both rich sources of hemicellulose and lignin. With all probability hemicellulose and lignin are not completely eliminated with the traditional processing, and thus they appear in the spectra.

In fact, the two kinds of paper can be easily distinguished. However, when Japanese paper is exposed to UV ageing, hemicellulose and lignin spectral features disappear, making the papers different only in terms of the water absorption. This behaviour is consistent with previous works suggesting that residual hemicellulose, pectin, and lignin could act as a sacrificial substrate during irradiation [18–20]. The composition of the potash cooked mulberry fibres largely preserved the cellulose from oxidation, making Japanese paper which is known for its excellent stability and permanence [93]. Wood exposure to accelerated ageing is helpful in explaining the variation in the intensity of the peak at around 1730 cm$^{-1}$ (Figure 6b). After UV ageing, the peak at 1507 cm$^{-1}$ decreases in intensity, while the peak at 1730 cm$^{-1}$ has increased. The formation of degradation products [16,19] evidently affects the intensity of the higher peak.

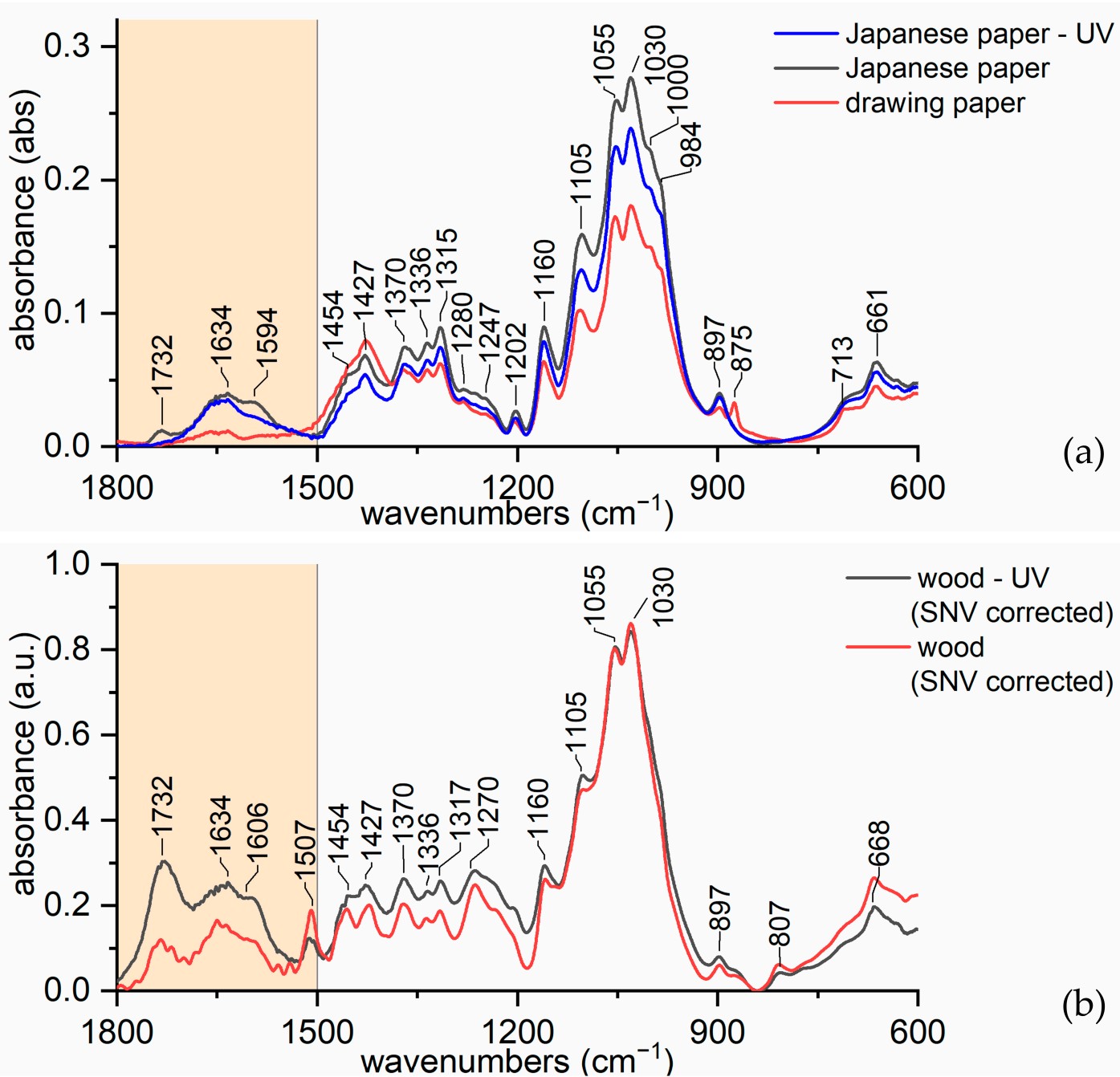

**Figure 6.** (**a**) Detail (region 1800–600 cm⁻¹) of ATR-FTIR spectra of commercial drawing paper and Japanese paper before and after UV ageing for 60 h; (**b**) Detail of ATR-FTIR spectra of wood before and after UV ageing for 60 h.

Another issue in terms of differentiating between compounds is the water content that the fibre can adsorb. This is strictly related to the relative amounts of crystalline and amorphous phases. Peaks at 1600 cm⁻¹ (pectin), and 1606, 1595 cm⁻¹ (lignin) can be concealed or enhanced by the water absorption band. In order to study its contribution, cotton samples were stored in three different conditions of relative humidity.

Figure 7a shows the whole spectra, where the most evident fact is that at high humidity levels, the general intensity of spectrum is enhanced. The region 1750–1500 cm⁻¹ contains only the contribution from water OH bending. After 65 h at 40 °C, the contribution has almost disappeared. The only water that is still bound is part of the freezing water. The wide band centred at 3300 cm⁻¹ is thus reduced.

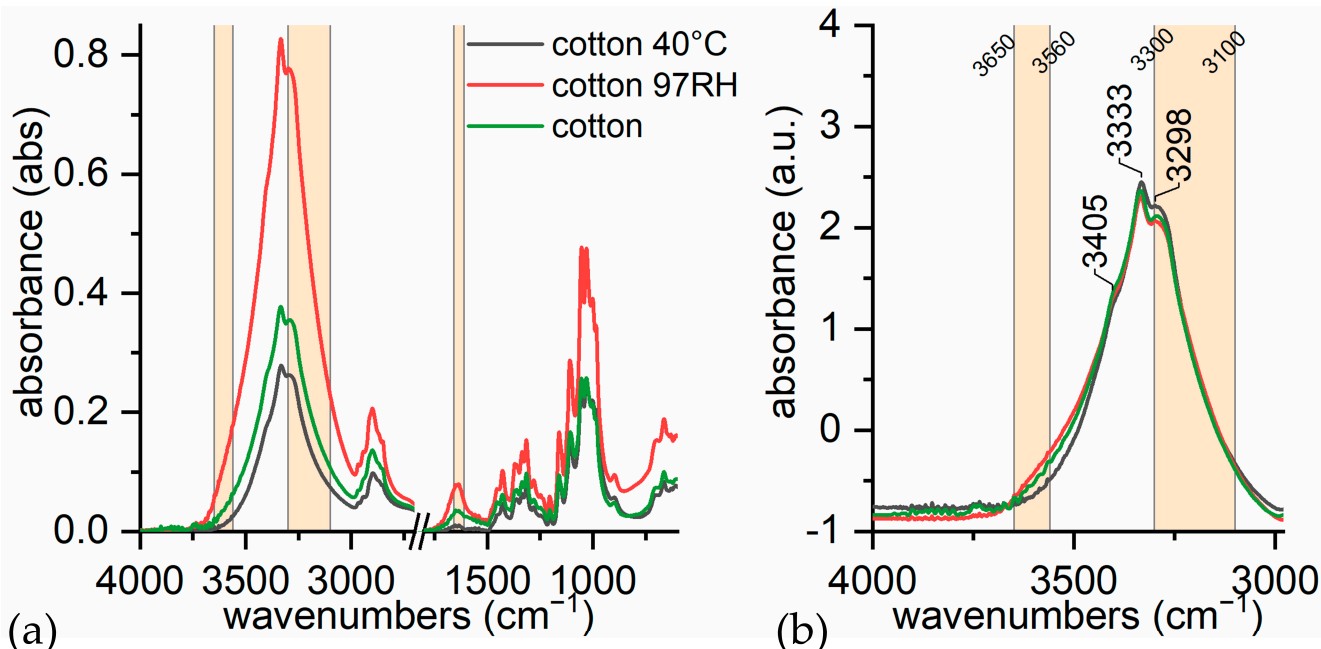

**Figure 7.** (**a**) ATR-FTIR spectra of cotton samples under different relative humidity conditions; (**b**) detail of the 4000–3000 cm$^{-1}$ region (SNV treated spectra). Coloured regions highlight where water stretching and bending contributions are located.

Figure 7b shows only the region 4000–3000 cm$^{-1}$, to which SNV was applied. It can be easily visualised that:

(i)  in dry conditions, the contributions from loosely bound water (3580 and 3555 cm$^{-1}$) are much lower;

(ii)  at room humidity, contributions from water absorption are higher, both at 1640 cm$^{-1}$ and at 3300 cm$^{-1}$;

(iii)  under high humidity conditions (RH 97% for 65 h), the peak at 1640 cm$^{-1}$ is higher and more resolute (1636 cm$^{-1}$). The band at 3300 cm$^{-1}$ is not very different from the sample at room humidity, while the contribution at 3200 cm$^{-1}$ is slightly higher. This is in accordance with Berthold's work [94], which refers that uncharged cellulose does not contain freezing water even at RH 97%.

Note that the band at 3300 cm$^{-1}$ is made up of the superimposition of different contributions. Table 2 shows the different contributions from the water uptake.

**Table 2.** ATR-FTIR signals of water interaction with cellulose I (cotton) and II (viscose).

| Water $\tilde{\nu}$ (cm$^{-1}$) | Cotton $\tilde{\nu}$ (cm$^{-1}$) | Viscose $\tilde{\nu}$ (cm$^{-1}$) | Assignment |
|---|---|---|---|
| 3600–3560 | | | ν OH, loosely bound water [20] |
| | | 3484, sh | ν O(6)H, free or weakly bound (primary alcohol) |
| | | 3440, sh | ν O(2)H, free or weakly bound (secondary alcohol) [45] |
| | 3460–3410 | | O(2)H· · · · · ·O(6), intramolecular [44,45,47,49] |
| | 3375–3340 | 3356 | O(3)H· · · · · ·O(5), intramolecular [44,45,47,49] |

**Table 2.** *Cont.*

| Water $\tilde{\nu}$ (cm$^{-1}$) | Cotton $\tilde{\nu}$ (cm$^{-1}$) | Viscose $\tilde{\nu}$ (cm$^{-1}$) | Assignment |
|---|---|---|---|
| | 3310–3230 | | O(6)H· · · · · ·O(3′), intermolecular [44,45,47,49] |
| 3300–3100 | | | ν OH, strongly bound water [20] |
| | | 3151 | O(2)H· · · · · ·O(2′) and/or O(6)H· · · · · ·O(2′), intramolecular [44,45,47,49] |
| 1643–1634 | | | δ OH of adsorbed water [37,56,57,63] |

### 3.2. Discriminating Natural and Modified Cellulose Fibres

Figure 8 shows different regions of ATR-FTIR spectra of reference cotton and viscose. As summarised by the literature [56,57], peak shifts appear in the region 1800–600 cm$^{-1}$. In our spectra, the main differences are located at 1456, 1204, 1161, 1031, and 983 cm$^{-1}$, which shift to 1467, 1198, 1155, 1018, and 994 cm$^{-1}$ in viscose, respectively. Some extra peaks which are not reported also appear in our viscose reference spectrum. In contrast, peaks of cotton at 1428 and 1248 cm$^{-1}$ do not appear in viscose spectrum, as the latter is typical of hemicellulose which is not present in viscose. Different H bonding interactions could be responsible for some shifts in bending vibrations of O-H groups (1456 >> 1467, 1236 >> 1228, 1204 >> 1198 cm$^{-1}$) [57]. Similarly, stretching vibrations of C-O bonds may be influenced by the formation of different intramolecular and intermolecular H bonding. In fact, spectral differences can be observed, such as different intensities (1054 cm$^{-1}$, which is attributable to C(3)-O), and shifts (1031 >> 1019 cm$^{-1}$, arising from C(6)-O in conformation I [59]).

Band assignation in the 1000–920 cm$^{-1}$ region is difficult, both due to the coupled form of vibrations (glycosidic bond, C-OH stretching, ring vibrations) and an insufficient resolution of the majority of the bands [84]. If we assume that the peak at around 1000 cm$^{-1}$ is assigned to the glycosidic bond, the small variation (1001 >> 998 cm$^{-1}$) is explained as follows. The change in macromolecule chain conformation at the transition from one polymorphic form of cellulose I to II causes variations in the dihedral angles at the glycosidic linkage [84]. Other studies have assigned the shift to variations in the hydrogen bonding scheme, which involves ν C(6)-O in conformation II [57,59]. The peak at 984 cm$^{-1}$, which only appears in cotton spectrum, can be attributed to hemicellulose.

These minor differences may be misunderstood when the other molecules produce signals in this region. In our opinion, it is better to focus on the region between 3800 and 3000 cm$^{-1}$. Here, the hydroxyl band shows significant shifts, which are related to the change in crystalline structure [95]. It is unlikely that other compounds will show signals in this region. Assignments for cotton and viscose are presented and compared in Table 2.

According to Maréchal [59], the hydrogen bond network profoundly influences the OH stretching vibrations of alcoholic groups. As reported by Oh [57], this bonding scheme is almost destroyed when crystalline cells are transformed from cellulose I to II. The hydrogen-bonding pattern changes [57,59] are shown in Figure 1. Table 2 can thus be explained as follows.

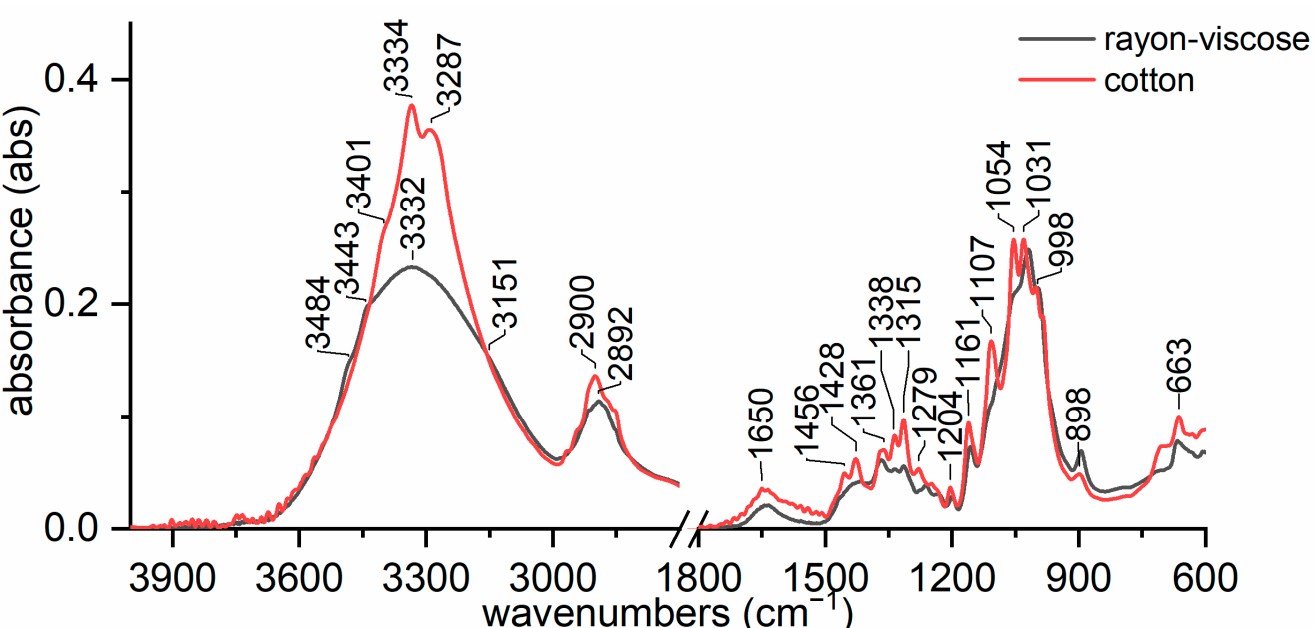

**Figure 8.** Detail of ATR-FTIR spectra of reference cotton and viscose.

The peak at around:

(i)   $3280\,\text{cm}^{-1}$ (O(6)H$\cdots\cdots$O(3′), intermolecular) disappears as it is replaced by $3151\,\text{cm}^{-1}$ (O(6)H$\cdots\cdots$O(2′), intramolecular) and $3480\,\text{cm}^{-1}$.

(ii)  $3340\,\text{cm}^{-1}$ (O(3)H$\cdots\cdots$O(5), intramolecular) does not change significantly (which also applies to $1055\,\text{cm}^{-1}$, which is the C(3)O stretching, and $1315\,\text{cm}^{-1}$, which is the O(3)H bending); the shift could be related to a higher water uptake [61].

(ii)  $3410\,\text{cm}^{-1}$ (O(2)H$\cdots\cdots$O(6), intramolecular) disappears as it is replaced by $3151\,\text{cm}^{-1}$ (O(2)H$\cdots\cdots$O(2′), intramolecular), 3450 and $3480\,\text{cm}^{-1}$.

(iv)  $3450\,\text{cm}^{-1}$ (O(2)H, secondary alcohol) is more evident because in viscose, the amount of water H bonded O(2)H is higher than in cotton.

(v)   $3480\,\text{cm}^{-1}$ (O(6)H, primary alcohol) is more evident because in viscose the number of water H bonded O(6)H is higher than in cotton.

The fifth peak can be understood as follows. When the O(2)H$\cdots\cdots$O(6) bond is present, C(6)O stretching at $1035\,\text{cm}^{-1}$ corresponds to O(6)H stretching at $3410\,\text{cm}^{-1}$. As expected from primary alcohols, C-O stretching is inversely correlated to O-H stretching. As we assigned $1015\,\text{cm}^{-1}$ to C(6)O in viscose, O(6)H stretching is expected to shift to $3480\,\text{cm}^{-1}$, as observed experimentally. According to Cichosz [61], in cellulose I, water molecules are mainly bound to O(3) and O(5), but in viscose also O(2) probably offer free sites to H bonding [62].

In particular, O(6) does not appear to be involved in water binding [63]; in fact, no peaks appear that could be assigned to that stretching. In contrast, in cellulose II, this vibration appears, together with that at $3450\,\text{cm}^{-1}$; which in our opinion, can be attributed to water bonded O(2) and O(6).

Figure 9a,b report a tentative deconvolution of the O-H stretching band for cotton and viscose, respectively. The contribution of the intramolecular bond O(3)H$\cdots\cdots$O(5) is much higher in viscose than in cotton, as is the contribution of strongly bound water. They could be correlated, as the less crystalline structure of viscose enables water to penetrate and interact deeply with cellulose. Similarly, the higher wavenumber is probably related to bond shortening, which has also been reported in cellulose I under wet conditions [61]. In both cases, the contribution from loosely bound water, i.e., water indirectly bonded to the OH groups via another water molecule, has little importance. This is in accordance with Berthold's quantification of freezing water [94].

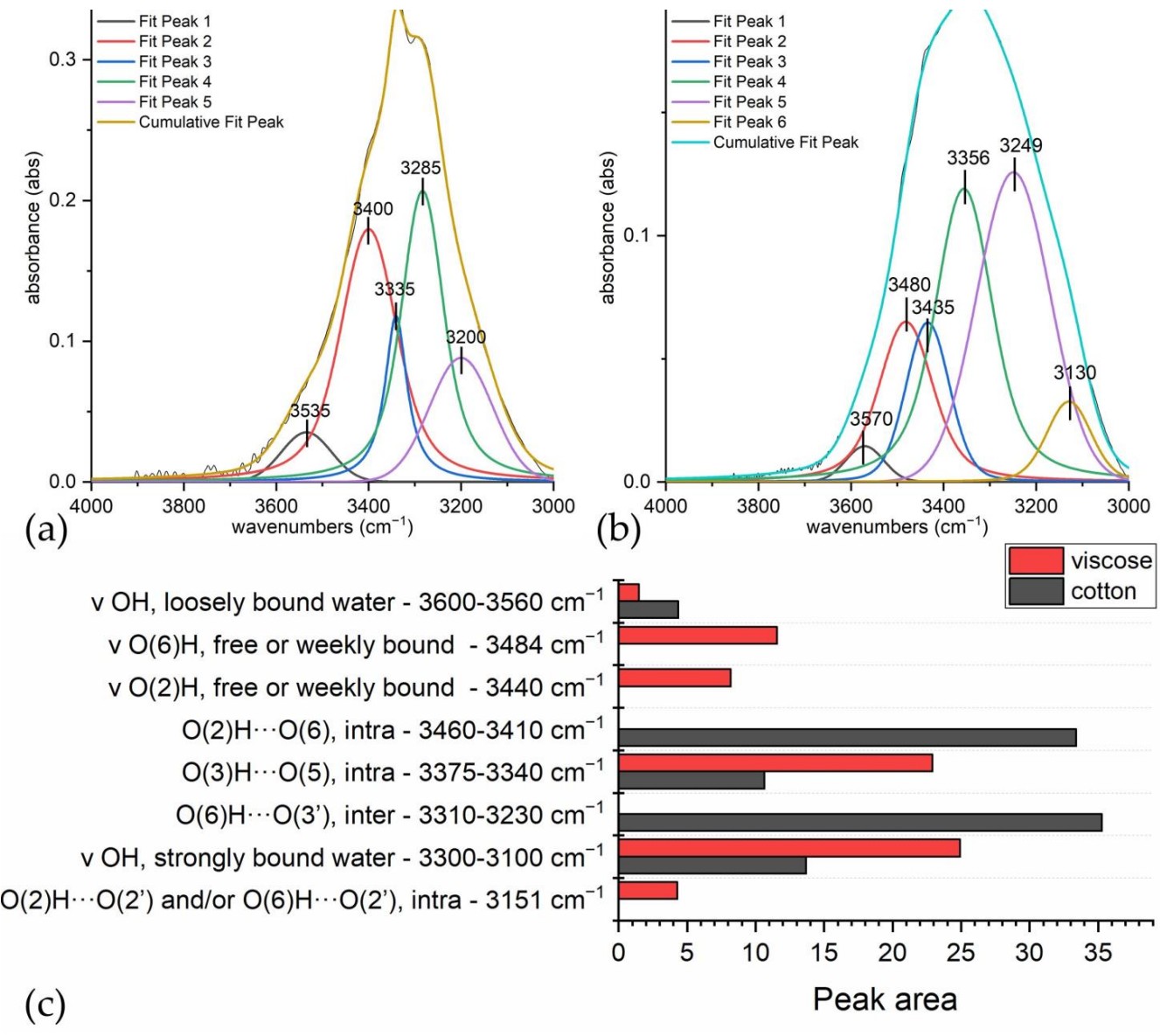

**Figure 9.** Deconvolution of the O-H stretching band: (**a**) cotton; (**b**) viscose; (**c**) comparison of the peak areas for each bond in cotton and in viscose.

In order to show that environmental humidity does not affect such spectral differences between cotton and viscose, Figure 10 shows ATR-FTIR spectra of viscose samples which were stored under three different conditions of relative humidity.

The whole spectra (Figure 10a) shows that no shifts take place due to changes in environmental humidity. The water uptake leads to increased levels of diffused intensity. By applying SNV processing to the spectral region between 4000 and 3000 cm$^{-1}$ (Figure 10b), strictly bound water is not desorbed from viscose during the heating cycle, but loosely bound water is. In high humidity conditions, viscose swells and is saturated with water, which is manifested by a band broadening in both strictly and loosely bound states.

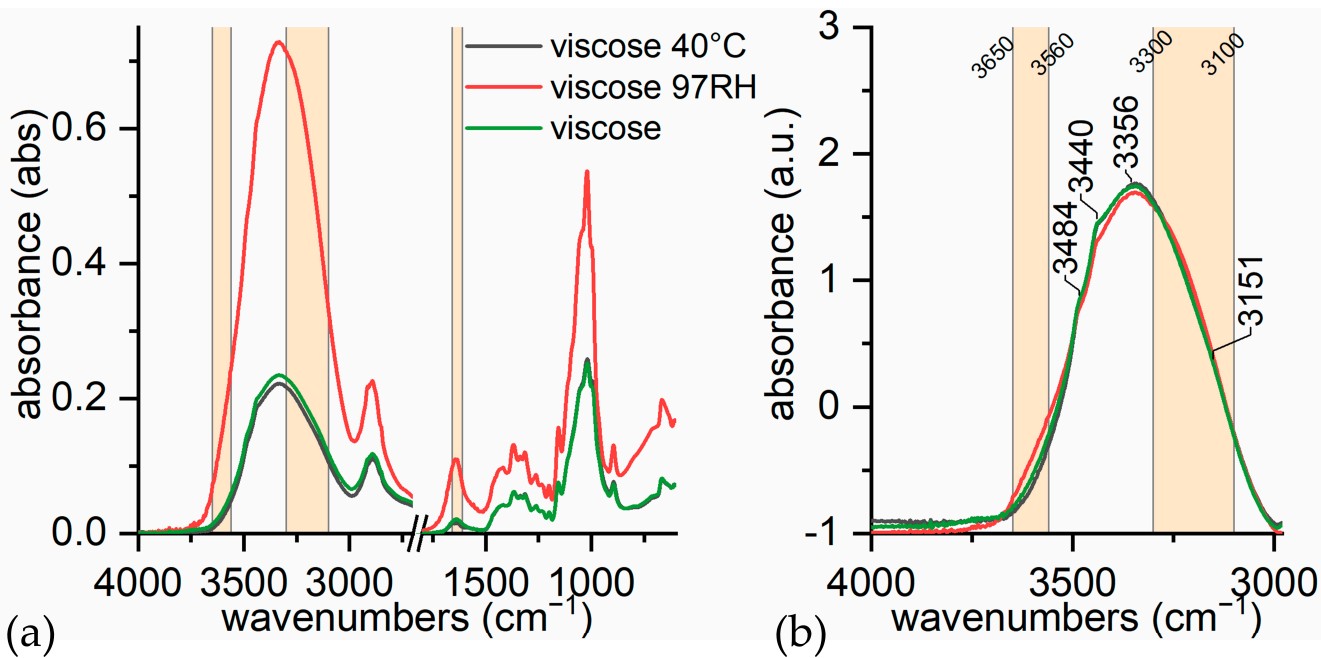

**Figure 10.** (**a**) ATR-FTIR spectra of viscose samples under different relative humidity conditions; (**b**) detail of the 4000–3000 cm$^{-1}$ region (SNV treated spectra). Coloured regions highlight where water stretching and bending contributions are located.

*3.3. Case Study: Traditional Samurai Armours*

Samples 1_11, 5_3, 7_6a, 7_6b, 8_1, and 8_9 were observed with SEM. Sample 1_11 is shown in Figure 11a; others are not shown, but the results for all samples are summarised in Table 3. Cotton was easily recognised according to the main features which were summarised in Section 3.1. Similarly, sample 8_1 was recognised as a bast fibre, but the recognition of bast fibre type is difficult with SEM observation [36]. Similarly, sample 5_3 was recognised as paper. Specific regions of the ATR-FTIR spectra are reported in Figure 11e,f, while the whole spectra are available in Figure S3.

**Table 3.** Results of analyses of samples from Morigi collection.

| Sample | Recognised Material (by Microscopy) | Recognised Material (by ATR-FTIR) |
| --- | --- | --- |
| 1_11 | cotton | cellulosic (no lignin) |
| 5_3 | paper | cellulosic (with lignin) |
| 6_1 | synthetic | rayon-viscose |
| 7_6a | cotton | cellulosic (no lignin) |
| 7_6b | cotton | cellulosic (no lignin) |
| 8_1 | bast fibre | cellulosic (no lignin) |
| 8_9 | cotton | cellulosic (no lignin) |
| 9_6 | swelled cotton | mercerised cotton (NaOH) |
| 10_1 | swelled cotton | mercerised cotton (NH$_3$) |

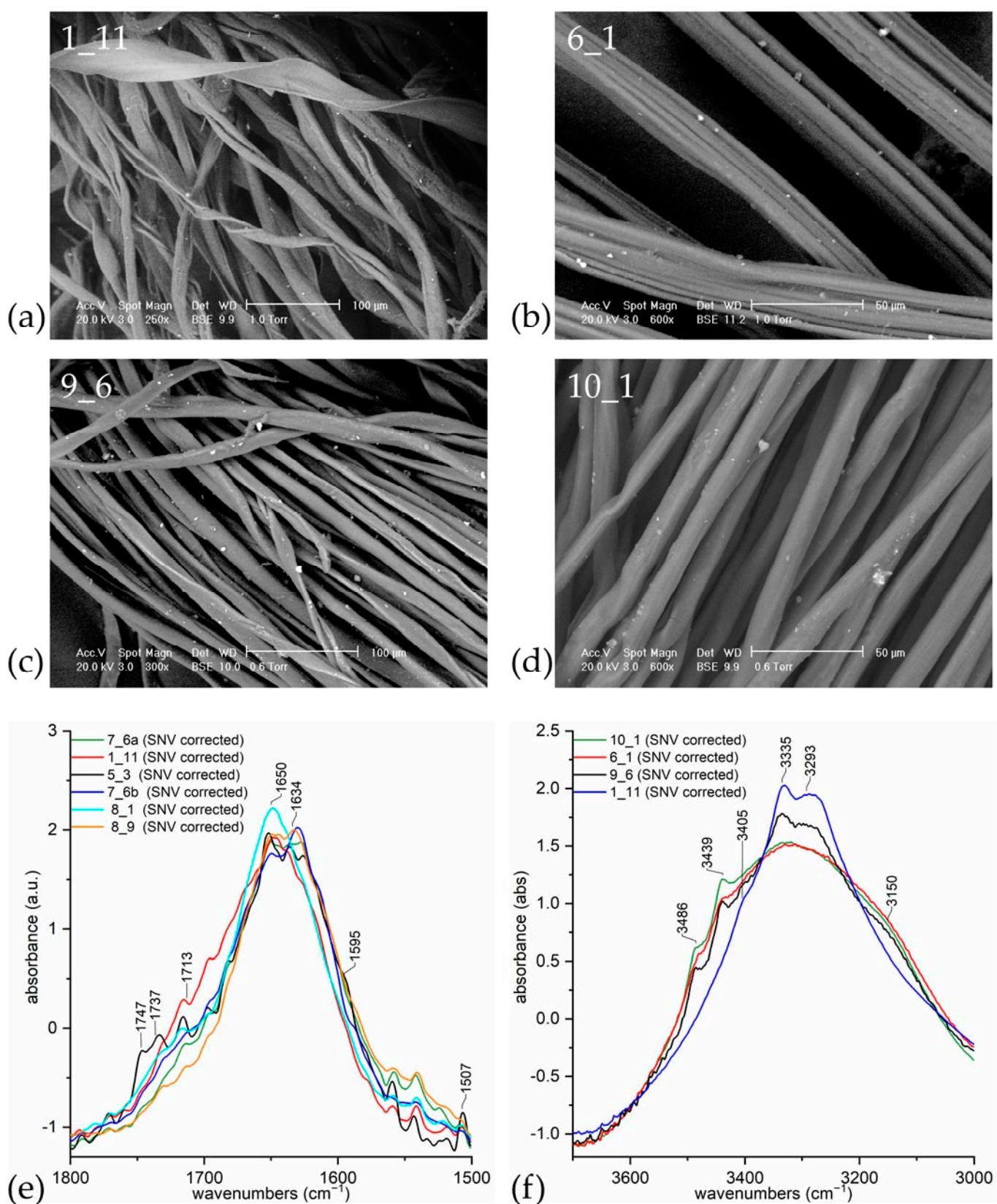

**Figure 11.** SEM images of samples 6_1 (**b**), 9_6 (**c**), and 10_1 (**d**), together with reference cotton sample 1_11 (**a**). (**e**) Detail (region 1800–1500 cm$^{-1}$) of ATR-IR spectra of samples 1_11, 5_3, 7_6a, 7_6b, 8_1, 8_9. (**f**) Detail (region 3700–3000 cm$^{-1}$) of ATR-IR spectra of samples 6_1, 9_6 and 10_1, together with the spectrum of cotton reference 1_11.

No information could be obtained from the whole spectra, except that they were all constituted by cellulosic fibres. The region 1800–1500 cm$^{-1}$ (Figure 11e) showed that sample 5_3 probably contained lignin and pectin (peaks at 1747, 1737, 1713, 1595, 1507 cm$^{-1}$). In fact, it was paper. Sample 8_1 was made of a bast fibre, which was identified through the SEM image, as its ATR spectrum was not dissimilar from other ones made of cotton. Similarly, it is not clear whether variability in the band at around 1650 cm$^{-1}$ was attributable to signals from a higher content of oxidation products (such as conjugated ketones [16]) or from the bending of water. The importance of SEM observation in order to obtain a clear identification is evident. A sequential procedure to differentiate cellulose fibres is schematised in Figure 4.

Some samples (6_1, 9_6 and 10_1) were not easily identified by OM and SEM images. A comparison of their SEM images (Figure 11b–d), revealed a great variability which was unexpected. Compared to cotton fibre (Figure 11a), other fibres appeared to be swollen (Figure 11c) or very swollen (Figure 11d). Fibres of sample 6_1 showed a completely different morphology due to the uniformity and longitudinal striations which are typical of artificial fibres (Figure 11b). In a previous work [96] we identified sample 6_1 fibres as viscose.

In order to explain this variability, we analysed the FTIR spectra (Figure 11f). While most of the ATR spectrum was in accordance with a cellulosic fibre (Figure S3), different signals were highlighted in the OH stretching band.

As mentioned previously, peaks at 3486 and 3439 cm$^{-1}$ and the absence of the peaks at 3405, 3335, and 3293 cm$^{-1}$ were spectral features that are also typical of viscose. In fact, cellulose I, which is characteristic of cotton and other natural fibres, turns into cellulose II when treated with a specific concentration of an aqueous NaOH solution ($\approx$10 wt%) [57,97]. Both viscose production and cotton mercerisation require the use of an NaOH solution, thus viscose and mercerised cotton possess a structure based on cellulose II, at least in part [56,57,98,98].

Sample 6_1 showed signals of cellulose II, while sample 9_6 showed both the spectral features of cellulose I and II, thus suggesting that the treatment with soda was partial, as happens in the production of mercerised cotton.

The analysis of the morphology (Figure 11b–d), led to other considerations. While sample 6_1 possessed the typical morphology of viscose fibre, sample 9_6 resembled cotton, but with a more swollen appearance. The changes observed in fibre morphology by mercerisation [98,99] (including deconvolution, a decrease in the size of the lumen, and a more circular cross section) appeared accordingly. In contrast, sample 10_1 showed a different morphology, which could be attributed to cotton treated with liquid ammonia [100], or to a regenerated fibre such as lyocell [101]. However, cotton treated with liquid ammonia, which turns from cellulose I to III, should only show the peak at 3484 cm$^{-1}$ [56,100], while the peak at 3439 cm$^{-1}$ also appeared. Lyocell was thus a better candidate, showing both peaks [102,103] and an appropriate morphology, however its production started in 1992 (commercialised as Tencel). This hypothesis suggests that the sample is a non-original material. All the results are summarised in Table 3.

## 4. Conclusions

The aim of this paper was to demonstrate that ATR-FTIR spectroscopy can be used to differentiate between natural and regenerated cellulosic fibres, also in historical samples. In particular, the OH stretching was found to be the most diagnostic region. The method is quick and simple to use during an analytical campaign on a textile collection and enables both historical natural and recent regenerated fibres to be identified from a microsample. Mercerised fibres can also be identified.

We believe that this kind of information can help to reveal past restoration materials and to reconstruct the history of the work of art, both important issues for informed conservation [32]. As modified cellulose fibres appeared in the 20th century, the identification of the chemical modifications taking place in the fibre gives the earliest date possible or clear

evidence of a recent restoration. This knowledge could help in deciding the best conditions in which to display objects and to stabilise them for long-term storage.

ATR-FTIR potentiality in differentiating natural fibres was also tested. A complete analysis of the FTIR band assignment of the main components in natural fibres was carried out in order to highlight the differences and to reveal the diagnostic bands that differentiate the fibres from different plants. We found that the spectral differences highlighted can help in the recognition of the fibre.

(i)     As FTIR is highly sensitive to lignin, it could be considered a non-destructive alternative for Herzberg stain test, for the evaluation of the lignin content (paper).

(ii)    The lignin content can distinguish jute and paper, which are richer in it, from hemp, showing a lower content and from cotton and flax, which contains negligible amount of it.

(iii)   Wax content can be useful to distinguish hemp from other fibres, unless fibre processing eliminated wax content.

(iv)    Hemicellulose can distinguish flax from other fibres, unless fibre ageing eliminated hemicellulose content.

However, it was also shown that the decay tends to conceal these variances, as well as some problems may come from the presence of contaminants. The processing of the fibre can cause variability, too.

Thus, the observation with SEM or optical microscopy is always advisable and we consider it complementary to ATR-FTIR analysis. Although optical microscopy is cheaper and easier to use, SEM assures the best resolution, depth of field, and contrast. These characteristics are particularly beneficial when the fibres under observation are degraded, as in the case of archaeological remains. A sequential procedure to differentiate cellulose fibres using SEM coupled with ATR-FTIR is schematised in Figure 4.

Finally, water adsorption of the cotton and the viscose was investigated both by a complete bibliographical search of the band assignation for cellulose I and II in the region of OH stretching and by studying their ATR-FTIR spectra under low and high humidity conditions. We showed that water adsorption does not significantly modify the peak positions. The 3800–3000 $cm^{-1}$ region thus contained diagnostic peaks which can reliably discriminate between natural cellulosic fibres and rayon-viscose and mercerised cotton.

**Supplementary Materials:** The following supporting information can be downloaded at: https://www.mdpi.com/article/10.3390/heritage5040213/s1, Table S1. Band assignment of cellulose and of correlated compounds (water, lignin, pectin, hemicellulose, wax). Figure S1. The pictures of the whole historical objects. Figure S2. Whole ATR-FTIR spectra of reference materials of jute, hemp, flax (linen), and cotton. Figure S3. ATR-FTIR spectra of historical samples.

**Author Contributions:** Conceptualisation, L.G., M.L. and L.R.; investigation, C.C., L.G., S.M. and L.R.; writing—original draft preparation, C.C., L.G., M.L. and L.R.; writing—review and editing, F.P.C., C.C., L.G., M.L., S.M., L.R. and S.R. All authors have read and agreed to the published version of the manuscript.

**Funding:** This research received no specific grant from any funding agency in the public, commercial, or not-for-profit sectors.

**Institutional Review Board Statement:** Not applicable.

**Informed Consent Statement:** Not applicable.

**Data Availability Statement:** The data presented in this study are available on request from the corresponding author.

**Conflicts of Interest:** The authors declare no conflict of interest.

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
