# Peer review of "Differentiating between Natural and Modified Cellulosic Fibres Using ATR-FTIR Spectroscopy"

_heritage, doi:10.3390/heritage5040213_

Round 1

Reviewer 1 Report

I would like to thank the authors for this interesting contribution since there is a need for new data and experimentation on natural fibres and their identification via non- or micro-destructive analytical methods. The article provides relevant data to the field.

The article needs minor revisions. The article is clear, well-structured and relevant for research in the field of historical textile collection and particularly for the application of a diagnostic protocol for the study and investigation of natural and man-made fibres.

The article provides new evidence regarding the successful application of ATR-FTIR for identifying and characterising natural and man-made fibres. The detailed study and interpretation of the spectra obtained by ATR-FTIR are interesting, especially the assignation of potentially discriminating bands that advance the current knowledge in the field of research. One significant novelty is related to the investigation of water absorption of the fibres. However, I suggest providing more details regarding the investigated materials. It should be clearly stated that the successful application of ATR-FTIR on historical and modern natural and man-made fibres, as provided by the article results, does not necessarily mean that the same technique will give comparable results if applied to ancient/archaeological fibres as discussed by recently published articles (some of the literature references are already cited in the article). This aspect and the related issues must be improved with more argumentation and clarity in the introduction and the conclusion.

The authors can find detailed comments in the attached file.

Author Response

I would like to thank the authors for this interesting contribution since there is a need for new data and experimentation on natural fibres and their identification via non- or micro-destructive analytical methods. The article provides relevant data to the field.

The article needs minor revisions. The article is clear, well-structured and relevant for research in the field of historical textile collection and particularly for the application of a diagnostic protocol for the study and investigation of natural and man-made fibres.

The article provides new evidence regarding the successful application of ATR-FTIR for identifying and characterising natural and man-made fibres. The detailed study and interpretation of the spectra obtained by ATR-FTIR are interesting, especially the assignation of potentially discriminating bands that advance the current knowledge in the field of research. One significant novelty is related to the investigation of water absorption of the fibres. However, I suggest providing more details regarding the investigated materials. It should be clearly stated that the successful application of ATR-FTIR on historical and modern natural and man-made fibres, as provided by the article results, does not necessarily mean that the same technique will give comparable results if applied to ancient/archaeological fibres as discussed by recently published articles (some of the literature references are already cited in the article). This aspect and the related issues must be improved with more argumentation and clarity in the introduction and the conclusion.

We thank very much the reviewer for her/his words of appreciation. She/he grasped the meaning of our work and raised some issues to improve the overall quality of the paper. We modified the text accordingly.

We understand the reviewer’s doubt about the applicability of our results to different contexts. Considering that each case study has its own specific features and that it would be an error to use a method with an uncritical approach, one of our aims was just to show the variability which is associated with the analytical data and to point out that it is important to know problems which can occur.

Yet, we are confident that our results on the discrimination of natural and modified cellulose are quite reliable.

Comments

Abstract:

(line 22) the application of ATR-FTIR for identifying ancient and historical textile fibres is not widely used. The technique is more commonly used in the conservation field to investigate the fibres' degradation and their state of preservation.

The concept was rephrased, as follows (lines 22-24 of the revised manuscript):

This paper presents the limitations and potential of ATR-FTIR spectroscopy applied to the study of cellulosic textile collections. The technique helps to differentiate natural fibres according to the content of lignin, pectin, hemicellulose and wax, although some problematic issues should considered.

(lines 27-28) The difference between "natural yarn" and "natural fibres" is unclear. Please, rephrase the concept.

We considered the two terms as synonyms. The text was revised following the reviewer’s suggestion, using only the term fibre for clarity sake.

  1. Introduction:

The section is clear, and the literature references cited are appropriate. However, in the introduction, the part related to the past cultivation and use of vegetable fibres needs to be more detailed and appears lacking in literature (lines 39-42). The question related to the early cultivated plant (for example, flax or hemp) used to extract textile fibres in the ancient world is still debated. I suggest, at least, adding some references.

A wider historical introduction would be beyond our aims and a too long introduction section would not be advisable. Anyway, we added some text and some references, as follows (lines 41-46 of the revised manuscript):

Vegetable fibres for textile materials were first used in prehistoric times [1,2]. Hemp is probably the oldest cultivated plant, which was widespread from Southeast Asia to China, where it seems have been used since around 4500 BCE. Several places of origin of the plant have been proposed, but probably archaeological findings in different centres are indicators of the diffusion of the plant in the early human agriculture [3]. Similarly, flax was certainly cultivated by the 4000 BCE in the Egyptian area, although it seems to have originated in the Near East [4,5]. Lastly, the art of spinning cotton appeared in India from around 3000 BCE, but it also developed independently in Peru [4].  

(lines 122-123) SEM, LM or PLM microscopes can be easily used to discriminate textiles fibres; the major issues are related to the discrimination between different bast fibres (such as flex, hemp, ramie, nettle etc.). Cotton can be easily identified (this is also true for animal fibres, such as wool). Bast fibres are characterised by similar morphology, and the diagnostic elements for discrimination, frequently described in the literature, do not consider the high natural variability of these parameters. Since the article focuses on vegetable and man-made fibres, I would suggest specifying that mainly bast fibres can be easily confused.

We clarified the issue as follows (lines 131-136 of the revised manuscript):

When the fibres need to be clearly identified, several methods can be used [31,32]. Microscopy has traditionally been used to differentiate fibres, using both light microscopy (LM) [33,34], polarized light microscopy [12,33,35] and scanning electron microscopy (SEM) [33,34,36,37]. While for cotton, for example, the identification is quite simple, some fibres can be easily confused due to the high natural variability of descriptive parameters. This is true especially for bast fibres [36]. So, considerable experience is required [35].

(lines 124-125) Morphological analyses are routine techniques to identify and distinguish vegetable and animal fibres. For this type of investigation, microscopy must be considered as the less expensive and time-consuming technique. Moreover, from the methodological point of view, microscopy must be considered the first analytical step to characterise natural fibres. The FTIR is currently applied to evaluate the state of preservation and the decay of the fibres or for trying to distinguish between animal and vegetable origin of highly degraded fibres (such as those found in archaeological context). FTIR is also frequently applied to discriminate among different types of bast fibres or silk sources (such as Bombyx Mori silk or wild species). Please, add more information about this issue.

We clarified the issue as follows (lines 141-148 of the revised manuscript):

Today, Raman [34,39] and FTIR spectroscopy [34,40–43] are established tech-niques to identify the nature of the fibres, both cellulosic and proteinaceous, generally coupled with SEM observation. In particular, ATR-FTIR spectroscopy is used to give chemical information about archaeological textiles, whose morphology is often very decayed due to the biological attack [33,40,41,44–46]. ATR-FTIR analysis permits also to detect traces [15] of dyes, mordants, contaminants and dirt [47,48] such as gypsum, kaolin, and various organic materials which can adhere to historical textiles [45] and papers [49].

(lines 136-142) Please specify that the cited literature about experimentation on natural fibres using ATR-FTIR is related to modern material.

We clarified the issue as follows (lines 160-167 of the revised manuscript):

Similarly, good results are obtained when ATR-FTIR data are processed with the chemometrics method in order to cluster fresh fibres from different sources [33,53]. However, the best performances are obtained when advanced chemometric methods are applied [54]. FTIR investigations into flax [55] have provided very detailed results which can estimate, for example, how fine the flax fibre is. Similarly, ATR-FTIR spectroscopy is particularly efficient [56,57] in discriminating between cellulose I, II and III obtained by processing cotton with NaOH solution and liquified anhydrous NH3, respectively.

  1. Materials and Methods

2.1. Reference materials

The experimental design of the articles is appropriate to test the hypothesis. However, I recommend specifying how modern fibres were processed, mainly if the modern reference samples of vegetable fibres were naturally processed or not (lines 203-204). Differences in the spectra and the microscopic structure of the fibres can be due to different processing. In Figure 3, bast fibres show different appearances that can be attributed to different processing stages. This is a fundamental issue that can also be related to the variation in the final ATR-FTIR spectra. This issue must be considered and discussed.

We briefly considered this kind of variability, as the reviewer can see at in lines 341-342 of the original paper. According to the reviewer’s request, we made some research, and we clarified the issue, as follows (lines 439-444 of the revised manuscript):

In addition, the presence of contaminants [15,47,48] and the processing of the fibre [55] could have some influence on the spectra. It is known that FTIR signal is influenced by the treatment of natural fibres with different reagents [87], even if historical methods are not reported to use chemicals [5]. For example, alkalisation with sodium hydroxide removes materials, such as hemicellulose, lignin, wax and pectin, from the surface of fibre bundles [88].

  1. Results and Discussion

3.1. Variability of natural cellulose fibres

In general, images, figures and tables properly show the data. However, Figure 4 cited in line 329 seems wrongly refer to the SEM micrograph of flex fibres, whereas Figure 4 shows the ATR-FTIR spectra of cotton fibres (page 13 of the manuscript). Moreover, in addition to the microscopic images, I would suggest including pictures of the whole historical objects/specimens analysed in the article.

The reviewer misunderstood the sentence. Now it has been rephrased in a clearer way. The text was revised as follows (lines 424-426 of the revised manuscript):

As proof, in Figure 5 we report spectra of different historical samples of flax which were previously identified by SEM observations (not shown).

The pictures of the whole historical objects were added in the supplementary material (Figure S1).

  1. Conclusions

The conclusions are consistent with the data and arguments presented. However, I firmly believe that the word "ancient" (e.g. ancient fibres or ancient collection) should be avoided in this context (line 532). The materials analysed are not dated to antiquity since the oldest object dates to the 16th century. The issues related to the characterisation and investigation of archaeological or ancient organic fibres, which are exposed to much more severe degradation and alteration, are not proven or solved by the authors experimentations. More attention is required to use the word "ancient" throughout the article.

As the reviewer suggested, “ancient” was replaced by “historical”.

Reviewer 2 Report

The article “Discriminating natural and modified cellulosic fibres by ATR FTIR spectroscopy”, as the title suggests, is focused on the analysis of different fibres by means of ATR FTIR spectroscopy. The topic is interesting and relevant to the journal, and I would like to take this opportunity to compliment the authors for their work.

Overall, the results exhibit a very high quality, I found particularly interesting the idea of discriminating cotton and viscose by their water content.

However, I have few suggestions to improve the general quality of the paper.

I have noticed how the introduction is very complete in describing cellulose chemistry and the analytical techniques generally applied to the study of fibres. However, I would stress out more what is the need for an analytical protocol for fibres identification based solely on ATR-FTIR. The authors introduced how microscopy techniques are generally used for the characterization of natural fibres, providing generally satisfactory results. It is true that for accurate results microscopy analysis must be generally performed by well-trained experts, but it also requires very small samples, and it usually allows for the separation and identification of different kinds of fibres associated together. I have experience with FTIR analysis of fibres and in multiple cases I needed to manually separate morphologically dissimilar fibres under the optical microscope prior to performing FTIR analysis (in transmission).

More in detail:

Line 91 The most serious conservation problem for cellulosic materials is the photochemical damage [5,10], but oxidation can occur in wet and humid environment too [5,11,12].

While I understand what the authors are stating in this period, I find it confusing. I would specify that photochemical damage is the primary cause of oxidation, which can also occur in absence of light in humid environments. I would appreciate if the authors could explain what they mean with “wet” environment 

Lines 94-95 I would suggest mentioning the role of metal ions (often indeed contained also in mordants and dyes, but not only) in the oxidation of cellulose

Line 142. Please explain the term “liquid NH3”, I suppose it refers to ammonium hydroxide

Lines 179-182 The following sentence “Polar solvents like water possess a large possibility to form extra hydrogen bonds with the cellulose molecule, as it has less hydroxyl groups with protons capable of forming hydrogen bonds than the number of oxygen atoms that would like to form hydrogen bonds” is confusing. Please rephrase it in a clearer way.

Lines 212-214 Please explain more in detail the fabrication of the homemade climatic chamber

In the materials and method, please indicate the kind of crystal used for ATR analysis. Also, why were the spectra acquired using only 32 scans? Unless the instrument used was equipped with an MCT liquid nitrogen cooled detector (not specified in the materials and methods), 32 scans might have been too low to reach a satisfactory S/N ratio.

Figure 1 is confusing; I suggest stacking the spectra.

Table 1 is confusing; I suggest working on the layout to make it more compact and accessible

I see that table 1 takes into account several publications as references, were the FTIR measurements included in those publications acquired with similar conditions as the one presented in this paper? I suppose the majority of the publications in literature are based on transmission FTIR, then how do the authors justify comparing these values to ATR measurements? Did the authors consider the possibility of shifting in wavenumbers due to non-homogeneous penetration depths in ATR when comparing the spectra with transmission ones?

Line 318 “Similarly, the peak at 1735 cm-1 should be assigned to C=O of ester in pectin and hemicellulose”, why not waxes? Also the peak is market as 1732 cm-1 both in figure 1 and in table 1. Please double check. It would be great to see the CH stretching in Figure 1, they would clarify the possible presence of waxy components.

In any case, I would be concerned by the possible anomalous presence of waxes and proteins as they could indicate contamination (waxes and proteins, as well as gypsum and calcium carbonate are the most commonly contaminants in cultural heritage materials).

Line 320, when reporting that hemp is the only fibre to contain absorptions due to waxes, did the authors consider the possibility of degreasing treatment for the other fibres? How many commercial samples of each fibre were analyzed before drawing these conclusions? Are hemp fibres analysis expected to show always signals due to waxy materials? Can the authors specify were they purchased the reference samples?

Different flax samples were analyzed by ATR-FTIR and compared, was the same step performed for all the other fibres? How many samples were analyzed?

Line 350. Typo in “heicellulose”

Lines 328-329. Can the authors add a brief description of the criteria used for SEM identification of flax fibres? I think it would make a great addition to the discussion of the results.

Moreover, how were the SEM conditions for the analysis selected? I see that the analysis was performed in low vacuum (I suppose to avoid the sputtering of the samples), but no elemental analysis was performed. I would argue that 20kV is a very high value, that could easily compromise the morphology of natural samples.

Line 529. Typo “in in”

Author Response

The article “Discriminating natural and modified cellulosic fibres by ATR FTIR spectroscopy”, as the title suggests, is focused on the analysis of different fibres by means of ATR FTIR spectroscopy. The topic is interesting and relevant to the journal, and I would like to take this opportunity to compliment the authors for their work.

Overall, the results exhibit a very high quality, I found particularly interesting the idea of discriminating cotton and viscose by their water content.

However, I have few suggestions to improve the general quality of the paper.

I have noticed how the introduction is very complete in describing cellulose chemistry and the analytical techniques generally applied to the study of fibres. However, I would stress out more what is the need for an analytical protocol for fibres identification based solely on ATR-FTIR. The authors introduced how microscopy techniques are generally used for the characterization of natural fibres, providing generally satisfactory results. It is true that for accurate results microscopy analysis must be generally performed by well-trained experts, but it also requires very small samples, and it usually allows for the separation and identification of different kinds of fibres associated together. I have experience with FTIR analysis of fibres and in multiple cases I needed to manually separate morphologically dissimilar fibres under the optical microscope prior to performing FTIR analysis (in transmission).

We thank very much the reviewer for his/her words of appreciation. He could grasp the meaning of our work and he raised some issues which have been solved improving the overall quality of the paper.

As for the reviewer’s general considerations, we are not sure that we've caught the suggestions. As we point out in the final statement of the original paper, “A visual inspection through SEM is relatively non-invasive and it is still often decisive in recognizing natural fibres”. So, we are not proposing a protocol based only on ATR-FTIR spectroscopy. If it is probably the best way to discriminate among cotton, mercerized cotton and viscose, ATR-FTIR spectroscopy shows some limitations in the determination of other natural fibres. The technique can detect hemp, as it generally contains wax, and jute as it contains a lot of lignin, but some problems come with flax, which contains amounts of lignin too low to be detected, especially after ageing. In general, if one wants to extend FTIR analysis to a wide variety of bast fibres, a classification would be difficult. Basically, the technique can distinguish if the fibre contains lignin or not (similarly to Herzberg test), and to detect the presence of other components discussed in the paper. Thus, the observation with SEM or optical microscopy is always advisable and we consider it complementary to ATR-FTIR analysis. We clarified this point in the revised version of the manuscript, in the Conclusion.

In addition, we agree with you that the microscopic approach is fundamental to get a complete characterization when different fibres are associated together. However, ATR can suggest the presence of associated fibers, and mechanical separation can then be performed for reliable recognition. Besides, for the purpose of best conservation in museums, it is usually sufficient to know the main component of textiles, so we think that ATR-FTIR spectroscopy can give adequate results, given the advantages of fastness, easiness and non-invasiveness. We exposed other advantages of ATR-FTIR spectroscopy in the revised manuscript as follows:

(Line 168)

Finally, it is worth recalling advantages of analysis with ATR-FTIR spectroscopy:

- no sample preparation;

- no band saturation phenomena;

- time and cost saving;

- few micrograms or less are generally necessary;

- non-destructive, as the sample can be re-used for further investigations (although the pressure applied during the analysis can induce morphological modifications);

- extensive database available, as literature dealing with transmission FTIR can be generally extended to ATR spectra.

Line 91 The most serious conservation problem for cellulosic materials is the photochemical damage [5,10], but oxidation can occur in wet and humid environment too [5,11,12].

While I understand what the authors are stating in this period, I find it confusing. I would specify that photochemical damage is the primary cause of oxidation, which can also occur in absence of light in humid environments. I would appreciate if the authors could explain what they mean with “wet” environment

We rephrased the text and eliminated the term “wet”, which could be confusing. Some of the cited papers actually do tests with wet paper, but it is a very particular condition. We revised the text, as follows (lines 98-100 of the revised manuscript):

The most serious conservation problem for cellulosic materials is the photochemical damage [5,10], however oxidation can occur in dark and humid environment, especially when there are temperature fluctuations [5,11,12].

Lines 94-95 I would suggest mentioning the role of metal ions (often indeed contained also in mordants and dyes, but not only) in the oxidation of cellulose

We revised the text, as follows (lines 101-104 of the revised manuscript):

Reactive oxygen species (ROS) are produced by photochemical reactions, which are catalysed by the presence of transition metal ions - generally coming from mordants and dyes - and are accelerated by other factors such as moisture.

Line 142. Please explain the term “liquid NH3”, I suppose it refers to ammonium hydroxide

In the industrial process of mercerization, “the swelling agents generally employed are aqueous solutions of NaOH and liquefied anhydrous ammonia (L-NH3), with the former representing the majority case” (Manian AP, Braun DE, Široká B, Bechtold T. Distinguishing liquid ammonia from sodium hydroxide mercerization in cotton textiles. Cellulose. Springer Netherlands; 2022;29:4183–202). In order to avoid misunderstandings, we changed “liquid” into “liquified anhydrous”.

Lines 179-182 The following sentence “Polar solvents like water possess a large possibility to form extra hydrogen bonds with the cellulose molecule, as it has less hydroxyl groups with protons capable of forming hydrogen bonds than the number of oxygen atoms that would like to form hydrogen bonds” is confusing. Please rephrase it in a clearer way.

We revise the text, as follows (lines 214-216 of the revised manuscript):

Polar solvents like water can easily form extra hydrogen bonds with cellulose, as some oxygen atoms in the chain are non-hydrogen bonded proton acceptor [46] (Figure 1).

Lines 212-214 Please explain more in detail the fabrication of the homemade climatic chamber

We added a detailed description, as follows (lines 248-250 of the revised manuscript):

High humidity conditions were obtained by storing samples for 65 hours in a desiccator over K2SO4 salt, where RH level was monitored in situ by placing moisture data logger and set to around 97%.

We preferred to avoid referring to a climatic chamber anymore, as it is generally considered as a system which can control both humidity and temperature, at least. In this work, the temperature was not controlled.

In the materials and method, please indicate the kind of crystal used for ATR analysis. Also, why were the spectra acquired using only 32 scans? Unless the instrument used was equipped with an MCT liquid nitrogen cooled detector (not specified in the materials and methods), 32 scans might have been too low to reach a satisfactory S/N ratio.

We revise the text, as follows (lines 298-301 of the revised manuscript):

The detector is a fast recovery deuterated triglycine sulfate (DTGS). The analysing crystal is diamond, which shows typical absorption at around 2100 cm-1. For this reason, in figures the region 2400-1800 cm-1 is generally not shown.

The number of scans was found acceptable to obtain good quality spectra.

Figure 1 is confusing; I suggest stacking the spectra.

We changed it according to the reviewer’s suggestion. Please refer to Figure 3 in the revised manuscript.

Table 1 is confusing; I suggest working on the layout to make it more compact and accessible

We moved the table of assignments to the supplementary material (Table S1) without any changes, as we don’t know how to make it more compact without eliminating valuable information. Yet, according to the reviewer’s suggestion, we added a more compact signals table (Table 2) coupled with a flow chart (please refer to Figure 4 of the revised article) which should give guidelines to the reader to evaluate ATR spectra and SEM images.

I see that table 1 takes into account several publications as references, were the FTIR measurements included in those publications acquired with similar conditions as the one presented in this paper? I suppose the majority of the publications in literature are based on transmission FTIR, then how do the authors justify comparing these values to ATR measurements? Did the authors consider the possibility of shifting in wavenumbers due to non-homogeneous penetration depths in ATR when comparing the spectra with transmission ones?

Yes, we evaluated the issue, having experience with both transmission and ATR modalities. We know also specifical literature on the problem (Grdadolnik, Jože. "ATR-FTIR spectroscopy: Its advantage and limitations." Acta Chimica Slovenica 49.3 (2002): 631-642). For example, it is known that silk fibre shows a significant shift in the position of amide I peak (1640 cm-1 in transmission mode and 1620 cm-1 in ATR mode). In the case of cellulosic fibres, we didn’t notice the presence of shifts, as confirmed by the literature too (see for example Liu X, Renard CMGC, Bureau S, le Bourvellec C. Revisiting the contribution of ATR-FTIR spectroscopy to characterize plant cell wall polysaccharides. Carbohydr Polym. Elsevier Ltd; 2021;262).

Line 318 “Similarly, the peak at 1735 cm-1 should be assigned to C=O of ester in pectin and hemicellulose”, why not waxes? Also the peak is market as 1732 cm-1 both in figure 1 and in table 1. Please double check. It would be great to see the CH stretching in Figure 1, they would clarify the possible presence of waxy components.

Thank you for having noticed this error. We revised the text, as follows (lines 412-413 of the revised manuscript):

Jute contains much more lignin than the others [37], and thus shows clearer lignin peaks. Similarly, the peak at 1735 cm-1 should be assigned to C=O of ester in pectin, wax, and hemicellulose.

As for the CH stretching region, we don’t show it in Figure 3 (Figure 1 was a wrong numbering) as the region 1800-700 cm-1 would be too compressed. Anyway, we added the whole spectra in the supplementary material (Figure S2).

In any case, I would be concerned by the possible anomalous presence of waxes and proteins as they could indicate contamination (waxes and proteins, as well as gypsum and calcium carbonate are the most commonly contaminants in cultural heritage materials).

We are about to publish a paper on the analysis of silk textiles collections with ATR-FTIR spectroscopy, and we dedicated a whole section to the recognition of contaminants on textiles.

We clarified the issue, as follows:

(line 145)

ATR-FTIR analysis permits also to detect traces [15] of dyes, mordants, contaminants and dirt [47,48] such as gypsum, kaolin, and various organic materials which can adhere to historical textiles [45] and papers [49].

(line 439)

In addition, the presence of contaminants [15,47,48] and the processing of the fibre [55] could have some influence on the spectra.

Line 320, when reporting that hemp is the only fibre to contain absorptions due to waxes, did the authors consider the possibility of degreasing treatment for the other fibres? How many commercial samples of each fibre were analyzed before drawing these conclusions? Are hemp fibres analysis expected to show always signals due to waxy materials? Can the authors specify were they purchased the reference samples?

Different flax samples were analyzed by ATR-FTIR and compared, was the same step performed for all the other fibres? How many samples were analyzed?

We initially analysed some replicates of cotton and ATR-FTIR spectra showed no noticeable differences. On the contrary, replicates of flax showed some differences due to ageing. We concentrated on these fibres as they were more easily obtainable and because they are probably more commonly found in textiles dating back to the last centuries (Melelli, Alessia, et al. "Evolution of the ultrastructure and polysaccharide composition of flax fibres over time: When history meets science." Carbohydrate Polymers 291 (2022): 119584.).

As for hemp and jute, they are difficult to be found on the market. So, we consulted references to be reasonably sure that our reference samples were in accordance with literature. In particular, references always report that hemp shows strong wax signals. We agree that degreasing could theoretically conceal this evidence, so we recommend SEM as a complementary technique. We revise the text, as follows:

(line 416)

Hemp is the only one characterized by wax signals at around 1470, 730, 720 cm-1. However, wax signals can disappear due to the fibre processing [87,88].

(line 439)

In addition, the presence of contaminants [15,47,48] and the processing of the fibre [55] could have some influence on the spectra. It is known that FTIR signal is influenced by the treatment of natural fibres with different reagents [87], even if historical methods are not reported to use chemicals [5]. For example, alkalisation with sodium hydroxide removes materials, such as hemicellulose, lignin, wax and pectin, from the surface of fibre bundles [88]. Figure 6 shows an example.

We would like to point out that our research focuses on the discrimination of modified and natural fibres. So, the recognition of all cellulosic fibres was not the aim of the research and should be addressed in another research project that also considers the degreasing treatments. Of course, our work does not claim to be exhaustive, as more experimental work should be done.

Line 350. Typo in “heicellulose”

The text was revised.

Lines 328-329. Can the authors add a brief description of the criteria used for SEM identification of flax fibres? I think it would make a great addition to the discussion of the results.

Yes, we added a description of morphology of each fibres at lines 350-385.

Moreover, how were the SEM conditions for the analysis selected? I see that the analysis was performed in low vacuum (I suppose to avoid the sputtering of the samples), but no elemental analysis was performed. I would argue that 20kV is a very high value, that could easily compromise the morphology of natural samples.

We usually use SEM in low vacuum conditions and prefer not to golden samples to preserve fragments as is for other analyses and because gold is a contaminant in case of EDX analysis. In addition, we did EDX analysis in order to test the presence of contaminants on our reference samples, so high voltage was needed. Anyway, we verified that the voltage did not cause damage to the samples, as confirmed by literature (Bicchieri, Marina, et al. "Microscopic observations of paper and parchment: the archaeology of small objects." Heritage Science 7.1 (2019): 1-12.).

Line 529. Typo “in in”

The text was revised.

Reviewer 3 Report

The authors are proposing a protocol for analysis of cellulosic fibres by the mean of ATR-FTIR spectroscopy. The idea is highly interesting and the effort organizing the IR bands to different functional groups is to be very appreciated given its usefulness.  

But I do find problematic the way the authors build their article and their argumentation as a whole is not convincing, partly also due to a very complicated way of using English and excessive wording. I strongly suggest a language wash. 

Figure and table numbering has to be corrected figure 3 (in the caption it appears as figure 2) onward.

I strongly suggest figure 1 to have a better resolution since it is very difficult to read. 

Parts of the text are difficult to comprehend, as for example, the paragraphs 179-183.

In paragraph 192, "which can be easily eliminated by centrifuge or pressure" refers to which types of the listed waters?

Please, replace in Table 2 (1 in caption - correct, please) "Assignations" with the commonly used Assignments. 

Subchapter 3.2 is missing. Or is it again a wrong numbering?

The biggest problems I found to be in 3.1 part. The authors are referring to a lot of bands, but it is really difficult to retrieve them in the figures they appeal to. Are the band positions the ones from the spectra presented in the paper of the shifts cited in the literature? It was very difficult un get that. 

In the figure 8, it was very interesting to see the deconvolution of the water peak, but it would be very helpful if the authors could try to assign the deconvoluted bands. Their explanations from paragraphs 449-457 are very general and difficult to connect to the deconvoluted curves. And why only the cotton and viscose were chosen for deconvolution? 

In part 3.3, authors stated that 6 samples were observed under SEM (fig 10a-d), but only 4 pictures are listed. As well, in the figure 10f only four spectra are listed instead on six. Why? Th case study is not convincing, or the authors did not clearly point out what they wanted to prove. 

Author Response

The authors are proposing a protocol for analysis of cellulosic fibres by the mean of ATR-FTIR spectroscopy. The idea is highly interesting and the effort organizing the IR bands to different functional groups is to be very appreciated given its usefulness. 

We thank very much the reviewer for her/his words of appreciation. She/he will probably appreciate Figure 4 too, which we added at the suggestion of another reviewer. The flow chart should give guidelines to the reader to evaluate ATR spectra and SEM images.

But I do find problematic the way the authors build their article and their argumentation as a whole is not convincing, partly also due to a very complicated way of using English and excessive wording. I strongly suggest a language wash.

The text was corrected by a professional native-English speaking revisor, and we hope that the reviewer will appreciate the revised manuscript.

Figure and table numbering has to be corrected figure 3 (in the caption it appears as figure 2) onward.

The error was corrected.

I strongly suggest figure 1 to have a better resolution since it is very difficult to read.

We send to the editorial office TIFF images with higher quality.

Parts of the text are difficult to comprehend, as for example, the paragraphs 179-183.

We revise the text, as follows (lines 214-216 of the revised manuscript):

Polar solvents like water can easily form extra hydrogen bonds with cellulose, as some oxygen atoms in the chain are non-hydrogen bonded proton acceptor [46] (Figure 1).

In paragraph 192, "which can be easily eliminated by centrifuge or pressure" refers to which types of the listed waters?

We clarified the sentence, as follows (lines 225-227 of the revised manuscript):

According to the environmental humidity, three types of water can be found [10], showing different bond strengths: i) structural water; ii) bound water; iii) excess water. The latter can be easily eliminated by centrifuge or pressure.

Please, replace in Table 2 (1 in caption - correct, please) "Assignations" with the commonly used Assignments.

We corrected it.

Subchapter 3.2 is missing. Or is it again a wrong numbering?

It is not missing. You can find it at line 393 of the original version and at line 501 of the reviewed version.

The biggest problems I found to be in 3.1 part. The authors are referring to a lot of bands, but it is really difficult to retrieve them in the figures they appeal to. Are the band positions the ones from the spectra presented in the paper of the shifts cited in the literature? It was very difficult un get that.

The peaks which are reported in spectra are only indicative, as the figures have the only aim to show the pattern of the spectra. The peaks that are truly informative and useful for the purpose of the work are shown in Table 3 and discussed in the text. The spectra are very complex due to the number of peaks and showing all the interesting bands while maintaining the clarity of the spectrum is really difficult.

In the figure 8, it was very interesting to see the deconvolution of the water peak, but it would be very helpful if the authors could try to assign the deconvoluted bands. Their explanations from paragraphs 449-457 are very general and difficult to connect to the deconvoluted curves. And why only the cotton and viscose were chosen for deconvolution?

We actually made the assignments of the deconvoluted bands in Table 3 and in the text at lines 535-558, where the appearance /disappearance of each peak is discussed, and each peak is attributed to a definite H-bond. In order to make it clearer, we added the wavenumbers of the band maxima near to labels in Figure 9c.

Only cotton and viscose were chosen for deconvolution, as they represent natural and regenerated cellulosic fibres, respectively. All the other natural fibres we analysed are very similar to cotton as for the region 4000-3000 cm-1 and we considered that discussion of their deconvolution would be redundant. 

In part 3.3, authors stated that 6 samples were observed under SEM (fig 10a-d), but only 4 pictures are listed. As well, in the figure 10f only four spectra are listed instead on six. Why? Th case study is not convincing, or the authors did not clearly point out what they wanted to prove.

All the nine samples were carefully observed with SEM, but only the images relevant to the discussion were shown and discussed. General results are reported in Table 4. We clarified it in the revised version, as follows (lines 585-587 of the revised manuscript):

Samples 1_11, 5_3, 7_6a, 7_6b, 8_1, 8_9 were observed with SEM. Sample 1_11 is shown in Figure 11a, others are not shown, while the results for all samples are summarized in Table 4.

Likewise, Figure 11 shows the images which are important to catch the variability of cotton fibre. Slight morphological differences are associated to significant differences in ATR-FTIR spectra.

Reviewer 4 Report

This work is dealing with fiber discrimination by means of FTIR spectroscopy and SEM imaging. Following this, the authors present an extensive introduction, in which fiber chemistry is explained, together with the historic and up-to-today background regarding fibers used in textile and paper making. This is followed by an introduction to fiber studies’ background. I personally find this study of great importance, and I believe that it should be considered to be published in Heritage.

Two major points should be taken into consideration:

-        1. The thing that should be enhanced is the morphological description of each kind of fiber, as the authors seem to regard it as common knowledge.

-        2. It is some kind of relevant with the previous point. Both in the abstract, as well as in line 164 the authors claim to propose a protocol for fibers identification. The authors give extensive details regarding FTIR characteristics within the results and discussion section, and they somewhat summarize them in the conclusions. A reader -and myself to be honest- would expect to see a summary of this -which can be described as a proper “protocol”, either in the end of results and discussion, or in the conclusion, in the form of a list, a flow chart or a table, i.e. these are the morphological characteristics, and these are the spectral characteristics in each case. I strongly believe that this addition will make the proposed paper as a VERY useful tool for researchers.

Finally, the ATR-way of measurement advantages should be mentioned, such as analysis without any sample preparation. From personal experience, the only other way to study fibers through FTIR is with transmittance in micro-FTIR, where phenomena such as the fiber’s thickness result to band saturation.

Some minor points to take into consideration:

·       1.  Figure 2 (optical images of the samples). Most of the presented samples show coloring. Have the authors identified any dye bands in their collected spectra?

·        2. Lines 256-259. Please, mention the crystal your ATR accessory has. Considering that you have left out the 2400-1900 cm-1 region, I assume it is a diamond ATR.

·       3.  Lines 428-438 and the bar graph in page 16. The graph works perfectly to understand the differences between viscose and cotton, but please incorporate the peaks’ wavenumbers to make it perfect.

·        4. SEM images of fibers, in pages 8 and 18, optical images in page 8, and inside the manuscript it is reported that “they have been observed with sem, revealing their nature (lines 479-480)”, and “Further investigation on the type of bast fiber is not possible by means of SEM observation (line 486)”, and “Some samples (6_1, 9_6 and 10_1) were not easily identified by OM and SEM images. Actually, by comparing their SEM images … (lines 496-497)”. The SEM description and dissemination of the fibers is missing, in terms of morphological characteristics for each fiber type. I understand that the authors are mostly enhancing their FTIR results, but this analysis should be also present in the manuscript. In lines 497-501 and by the end of the results and discussion section (lines 503-517), the authors refer to different morphologies of the samples, but the starting point is missing, i.e. characteristics of every type.

Some minor language things to take into consideration:

·       1.  Whole document. Please, check the Journal’s terms regarding American or British English language, in the case of “fibres/fibers”

·       2.  Whole document. Please, check table and figure numbers, in captions and inside the document, as something is off.

·       3.  Line 138 “when best chemometric method are applied”

·        4. Line 263 “correction was corrected”

·        5. Line 264. “SNV” please, explain the abbreviation

·        6. Line 291 “which show”

·       7.  Lines 300-301. The caption should include the sem images.

·       8.  Line 317. Maybe “much more lignin”

·      9.  Line 355. “e”

·      10.  Line 381. “, the contribution has quite disappeared”

·        11. Line 388 “the contribute”

·       12.  Line 390 “different contribution”

Author Response

This work is dealing with fiber discrimination by means of FTIR spectroscopy and SEM imaging. Following this, the authors present an extensive introduction, in which fiber chemistry is explained, together with the historic and up-to-today background regarding fibers used in textile and paper making. This is followed by an introduction to fiber studies’ background. I personally find this study of great importance, and I believe that it should be considered to be published in Heritage.

We thank very much the reviewer for her/his words of appreciation. She/he could grasp the meaning of our work and raised some issues which have been solved improving the overall quality of the paper.

As for the major points, our answers follow.

  1. The thing that should be enhanced is the morphological description of each kind of fiber, as the authors seem to regard it as common knowledge.

We totally agree with you. We added a detailed morphological description of each reference sample (lines 350-385 of the revised manuscript):

  1. It is some kind of relevant with the previous point. Both in the abstract, as well as in line 164 the authors claim to propose a protocol for fibers identification. The authors give extensive details regarding FTIR characteristics within the results and discussion section, and they somewhat summarize them in the conclusions. A reader -and myself to be honest- would expect to see a summary of this -which can be described as a proper “protocol”, either in the end of results and discussion, or in the conclusion, in the form of a list, a flow chart or a table, i.e. these are the morphological characteristics, and these are the spectral characteristics in each case. I strongly believe that this addition will make the proposed paper as a VERY useful tool for researchers. Finally, the ATR-way of measurement advantages should be mentioned, such as analysis without any sample preparation. From personal experience, the only other way to study fibers through FTIR is with transmittance in micro-FTIR, where phenomena such as the fiber’s thickness result to band saturation.

One of the aims of our work is to propose guidelines for those without extensive ATR-IR spectroscopy expertise. As the reviewer suggested, it is not obvious that someone can do recognition through SEM observation, so we added some information about it. We added a flow chart and an easy-to-use table for differentiating signals from cellulose, lignin, pectin, hemicellulose and wax. Please refer to Figure 4 of the revised paper.

We pointed out the advantages of ATR-FTIR spectroscopy in the Introduction, as follows:

(Line 168)

Finally, it is worth recalling advantages of analysis with ATR-FTIR spectroscopy:

- no sample preparation;

- no band saturation phenomena;

- time and cost saving;

- few micrograms or less are generally necessary;

- non-destructive, as the sample can be re-used for further investigations (although the pressure applied during the analysis can induce morphological modifications);

- extensive database available, as literature dealing with transmission FTIR can be generally extended to ATR spectra.

As for minor points, our answers follow.

  1. Figure 2 (optical images of the samples). Most of the presented samples show coloring. Have the authors identified any dye bands in their collected spectra?

As shown by the literature, the dye identification with ATR-FTIR spectroscopy is difficult to achieve, mainly due to the highly intense signals from the substrate. Thus, articles on the issue are rare, also because there are techniques which are definitely more performing, for example SERS. Anyway, non-invasive infrared spectroscopy in reflection mode is a promising technique and are actually working on some of them. Results will soon be proposed in a publication, hopefully.

  1. Lines 256-259. Please, mention the crystal your ATR accessory has. Considering that you have left out the 2400-1900 cm-1 region, I assume it is a diamond ATR.

We added this information as follows:

(line 298)

The detector is a fast recovery deuterated triglycine sulfate (DTGS). The analysing crystal is diamond, which shows typical absorption at around 2100 cm-1. For this reason, in figures the region 2400-1800 cm-1 is generally not shown.

  1. Lines 428-438 and the bar graph in page 16. The graph works perfectly to understand the differences between viscose and cotton, but please incorporate the peaks’ wavenumbers to make it perfect.

We are happy that it works. We added peaks’ wavenumber in the Figure 9.

  1. SEM images of fibers, in pages 8 and 18, optical images in page 8, and inside the manuscript it is reported that “they have been observed with sem, revealing their nature (lines 479-480)”, and “Further investigation on the type of bast fiber is not possible by means of SEM observation (line 486)”, and “Some samples (6_1, 9_6 and 10_1) were not easily identified by OM and SEM images. Actually, by comparing their SEM images … (lines 496-497)”. The SEM description and dissemination of the fibers is missing, in terms of morphological characteristics for each fiber type. I understand that the authors are mostly enhancing their FTIR results, but this analysis should be also present in the manuscript. In lines 497-501 and by the end of the results and discussion section (lines 503-517), the authors refer to different morphologies of the samples, but the starting point is missing, i.e. characteristics of every type.

We thank you for the suggestion. We added the morphological description of the different fibres at the start of the discussion (lines 350-385 of the revised manuscript). Sentences across the text were rephrased according to the description.

As for minor language revisions, we corrected them across the text.

Round 2

Reviewer 2 Report

The article has greatly improved, once again I would like to compliment the authors on their work.

These are my comments on the revised version 

Line 143 ATR_FTIR

146 I am not sure about the “traces”, I would suggest “small amounts” or “low concentrations”.

Lines 149-151 the same concept is repeated twice “Generally speaking, FTIR can distinguish fibres from different cellulosic plants too, but problems arise when bast fibres are aged [33]. Unfortunately, it is difficult to distinguish fibres from different cellulosic plants, especially when bast fibres are aged [33].”

Line 222: “This is mainly due to the accessibility of the hydrogen bonds to moisture”.

I find this sentence confusing, can you rephrase it?

Line 228 “Structural and bound water penetrate cotton deeply, whereas they act as plasticizers for the fibre. As a consequence, regenerated fibres are more elastic than natural ones”

Do you have a reference to cite for the use of “plasticizer” in this context? Plasticizers are generally high boiling point chemicals added to polymers to modify their glass transition temperature and hence some physical properties

Line 416

How would the authors explain the high content of waxes in the hemp sample/s? Looking at the supplementary file and the CH stretching signals are indeed very sharp (which I agree with the authors could suggest the presence of waxes). The shoulder around 2952 cm-1 and the peak at 720 cm-1 (as well as the peak around 734 cm-1) remind me of beeswax, but it’s also true that candelilla and carnauba waxes also look very similar. I find this information very interesting. I agree that waxes can be generally present in high concentrations in hemp, up to 6.2% (Industrial Hemp Fibers: An Overview, https://doi.org/10.3390/fib7120106)

In any case, I apologize for my insistence but I am still not sure about the number of hemp samples that were analyzed and I also do not fully understand the author’s answer on this topic “As for hemp and jute, they are difficult to be found on the market. So, we consulted references to be reasonably sure that our reference samples were in accordance with literature. In particular, references always report that hemp shows strong wax signals.

Finally, I appreciate the authors clarifying the following concept “We would like to point out that our research focuses on the discrimination of modified and natural fibres. So, the recognition of all cellulosic fibres was not the aim of the research and should be addressed in another research project that also considers the degreasing treatments. Of course, our work does not claim to be exhaustive, as more experimental work should be done.

However, although the recognition of cellulosic fibres is not your primary goal, in the conclusion you affirm that “iii) Wax content can be useful to distinguish hemp from other fibres”.

Lines 668-670: Thus, the observation with SEM or optical microscopy is always advisable and we consider it complementary to ATR-FTIR analysis. A sequential procedure to differentiate cellulose fibres using SEM coupled with ATR-FTIR is schematised in Figure 4.

What are the main advantages in applying SEM rather than optical microscopy (or vice versa?). I am thinking as a conservator/student who reads your paper and wants to apply your protocol for fibres discrimination, how would I choose the right microscopy technique? Optical microscopy is way cheaper than SEM and it is something that can be easily performed in a conservation lab without requiring any special sample treatment.

Referring to the authors’ cover letter: “We usually use SEM in low vacuum conditions and prefer not to golden samples to preserve fragments as is for other analyses and because gold is a contaminant in case of EDX analysis”. I never considered gold as a contaminant for EDX analysis, especially when working with fibres but it is an interesting point of view.

Thank you for referencing Bicchieri et al publication where they state that at similar conditions, they “did not observe beam damage in most of the analysed spots”. However, I am truly curious to know how the authors retrieved the fibres after SEM analysis, as the adhesive used to prepare the samples on the stub is generally very difficult to remove and exhibit very strong FTIR signals (same problem would apply to Raman and chromatography). Perhaps the authors used a different sample preparation?

Author Response

We sincerely thank the revisor for his/her appreciation words. He/she clearly appears as an expert in this field. The reviewer can find our answers in attached file.

Reviewer 3 Report

The paper has sensibly improved. Only some comments in the attached pdf.

Given the extensive work using SEM, I am wondering if the title should not be changed in order to acknowledge that.  

Author Response

We thank the reviewer for his/her words of appreciation. Our answers to his/her issues follow.

Given the extensive work using SEM, I am wondering if the title should not be changed in order to acknowledge that.  

SEM is among the keywords, but not the focus of the work. Unfortunately, the title is enhancing the most innovative result of the work, and SEM is not a technique which assures a discrimination between natural and regenerated fibres. SEM is important for the recognition of natural fibres, but we should change the entire title to include it.

Line 151

“Unfortunately, it is difficult to distinguish fibres from different cellulosic plants, especially when bast fibres are aged [33].” Do the authors mean "herbaceous plants"?

The reviewer is right. The term was corrected according to your suggestion.

Line 223

“This is mainly due to the accessibility of the hydrogen bonds to moisture [64].” I am not sure what the authors mean. Could you rephrase?

We clarified the sentence, as follows (lines 221-223 of the revised manuscript):

However, it should be considered that the absorption of water is controlled by the accessibility of the hydrogen bonds to moisture, which strongly depends on crystallinity degree [64].

Line 228

Plasticizers render the fibers more plastic. Authors should double-check their statement about elasticity.

We clarified the sentence, as follows (lines 227-230 of the revised manuscript):

As a consequence, regenerated fibres are more plastic than natural ones, due to the increased water uptake. This fact makes handling wet viscose textiles quite dangerous because they readily stretch and lose their shape [15].

Other typos were corrected as suggested.